# SCORE-BASED SELF-SUPERVISED MRI DENOISING

**Jiachen Tu, Yaokun Shi & Fan Lam**
Department of Electrical and Computer Engineering
University of Illinois at Urbana-Champaign
Champaign, IL 61820, USA
`{jtu9, yaokuns2, fanlam1}@illinois.edu`

## ABSTRACT

Magnetic resonance imaging (MRI) is a powerful noninvasive diagnostic imaging tool that provides unparalleled soft tissue contrast and anatomical detail. Noise contamination, especially in accelerated and/or low-field acquisitions, can significantly degrade image quality and diagnostic accuracy. Supervised learning based denoising approaches have achieved impressive performance but require high signal-to-noise ratio (SNR) labels, which are often unavailable. Self-supervised learning holds promise to address the label scarcity issue, but existing self-supervised denoising methods tend to oversmooth fine spatial features and often yield inferior performance than supervised methods. We introduce Corruption2Self (C2S), a novel score-based self-supervised framework for MRI denoising. At the core of C2S is a generalized denoising score matching (GDSM) loss, which extends denoising score matching to work directly with noisy observations by modeling the conditional expectation of higher-SNR images given further corrupted observations. This allows the model to effectively learn denoising across multiple noise levels directly from noisy data. Additionally, we incorporate a reparameterization of noise levels to stabilize training and enhance convergence, and introduce a detail refinement extension to balance noise reduction with the preservation of fine spatial features. Moreover, C2S can be extended to multi-contrast denoising by leveraging complementary information across different MRI contrasts. We demonstrate that our method achieves state-of-the-art performance among self-supervised methods and competitive results compared to supervised counterparts across varying noise conditions and MRI contrasts on the M4Raw and fastMRI dataset. The project website is available at: `https://jiachentu.github.io/Corruption2Self-Self-Supervised-Denoising/`.

## 1 INTRODUCTION

Magnetic resonance imaging (MRI) is an invaluable noninvasive imaging modality that provides exceptional soft tissue contrast and anatomical detail, playing a crucial role in clinical diagnosis and research. However, the inherent sensitivity of MRI to noise, particularly in accelerated acquisitions and/or low-field settings, can impair diagnostic accuracy and subsequent computational analysis. Unlike natural images, MRI data often contains Rician or non-central chi-distributed noise in magnitude images. Denoising has become an important processing step in the MRI data analysis pipeline. Higher signal-to-noise ratios (SNR) enable better trade-offs for reduced scanning times or increased spatial resolution, both of which can improve patient experience and diagnostic accuracy. Traditional denoising methods, such as Non-Local Means (NLM) Froment (2014) and BM3D Mäkinen et al. (2020), often rely on handcrafted priors such as Gaussianity, self-similarity, or low-rankness. The performance of these methods is limited by the accuracy of their assumptions and often requires prior knowledge specific to the acquisition methods used. With the advent of deep learning,

supervised learning based denoising methods Zhang et al. (2017); Liang et al. (2021); Zamir et al. (2022) have demonstrated impressive performance by learning complex mappings from noisy inputs to clean targets. However, they rely heavily on the availability of high-quality, high-SNR "ground truth" data for training—a resource that is not always readily available or feasible to acquire in MRI. Acquiring such clean labels necessitates longer scan times, leading to increased costs and patient discomfort. This limitation underscores the need for self-supervised denoising algorithms, which remove or reduce the dependency on annotated datasets by leveraging self-generated pseudo-labels as supervisory signals. Moreover, supervised approaches often face generalization issues due to distribution shifts caused by differences in imaging instruments, protocols, and noise levels Xiang et al. (2023), limiting their utility in real-world clinical settings. Techniques such as Noise2Noise Lehtinen et al. (2018), Noise2Void Krull et al. (2019), Noise2Self Batson & Royer (2019), and their extensions Xie et al. (2020); Huang et al. (2021); Pang et al. (2021); Jang et al. (2024); Wang et al. (2023) have demonstrated the potential to learn effective denoising models without explicit clean targets. Recent innovations tailored for MRI, like Patch2Self Fadnavis et al. (2020) and Coil2Coil Park et al. (2022), further exploit domain-specific characteristics to enhance performance. However, these self-supervised methods often have limitations. For example, Noise2Noise requires repeated noisy measurements, which may not be practical. Methods like Noise2Void and Noise2Self rely on masking strategies that can limit the receptive field or introduce artifacts, potentially leading to oversmoothing and loss of fine details. Approaches such as Pfaff et al. (2023) and Park et al. (2022) require additional data processing steps, restricting their applicability. In this paper, we introduce Corruption2Self, a score-based self-supervised framework for MRI denoising. Building upon the principles of denoising score matching (DSM) Vincent (2011), we extend DSM to the ambient noise setting, where only noisy observations are available, enabling effective learning in practical MRI settings where high-SNR data are scarce or impractical to obtain. An overview of the C2S workflow is illustrated in Figure 1. Additionally, we incorporate a reparameterization of noise levels for a consistent coverage of the noise level range during training, leading to enhanced training stability and convergence. In medical imaging, the visual quality of the output and the preservation of diagnostically relevant features are often more critical than achieving high scores on standard metrics. To address this priority, a detail refinement extension is introduced to balance noise reduction with the preservation of fine spatial feature. Furthermore, we extend C2S to multi-contrast denoising by integrating data from multiple MRI contrasts. This approach leverages complementary information across contrasts, indicating the potential of C2S to better exploit the rich information available in multi-contrast MRI acquisitions.

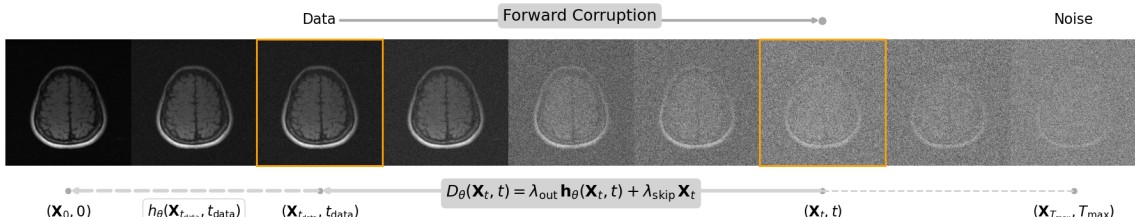

Figure 1: Overview of the Corruption2Self (C2S) workflow for MRI denoising. Starting from a noisy MRI image $\mathbf{X}_{t_{\text{data}}}$, the forward corruption process adds additional Gaussian noise to create progressively noisier versions $\mathbf{X}_t$. During training, the model learns to reverse this process by estimating the clean image $\mathbf{X}_0$ from these corrupted observations, despite having access only to noisy data. The denoising function $\mathbf{h}_\theta$ approximates the conditional expectation $\mathbb{E}[\mathbf{X}_0 \mid \mathbf{X}_t]$, effectively learning to denoise without clean targets. A reparameterized function $D_\theta(\mathbf{X}_t, t)$, which shares parameters with $\mathbf{h}_\theta$, is used to compute the loss.

We conduct extensive experiments on publicly available datasets, including a low-field MRI dataset, to evaluate the performance of C2S. Our results demonstrate that C2S achieves state-of-the-art performance

among self-supervised methods and, after extending to multi-contrast on the M4Raw dataset, shows state-of-the-art performance among both self-supervised and supervised methods. Notably, we are among the first to comprehensively analyze and compare self-supervised and supervised learning approaches in MRI denoising. Our findings reveal that C2S not only bridges the performance gap but also offers robust performance under varying noise conditions and MRI contrasts. This indicates the potential of self-supervised learning to achieve competitive performance with supervised approaches when the latter are trained on practically obtainable higher-SNR labels, particularly in scenarios where perfectly clean ground truth is unavailable, offering a practical and robust solution adaptable to broader clinical settings.

## 2 BACKGROUND

### 2.1 LEARNING-BASED DENOISING WITHOUT CLEAN TARGETS

A central concept in many of the self-supervised denoising methods is J-invariance Batson & Royer (2019), where the denoising function is designed to be invariant to certain subsets of pixel values. Techniques like Noise2Void Krull et al. (2019) and Noise2Self Batson & Royer (2019) utilize this property by masking pixels and predicting their values based on their surroundings, approximating supervised learning objectives without the need for clean data. Noise2Same Xie et al. (2020) extends these concepts by implicitly enforcing $\mathcal{J}$-invariance via optimizing a self-supervised upper bound. Approaches such as Neighbor2Neighbor Huang et al. (2021), Noisier2Noise Moran et al. (2020), and Recorrupted2Recorrupted Pang et al. (2021) create pairs of images from a single noisy observation to mimic the effect of supervised training objectives with independent noisy pairs. Noise2Score Kim & Ye (2021) exploits the assumption that noise follows an exponential family distribution and utilizes Tweedie's formula to obtain the posterior expectation of clean images using the estimated score function via the AR-DAE Lim et al. (2020) approach. This approach highlights a connection between Stein's Unbiased Risk Estimator (SURE)-based denoising methods Soltanayev & Chun (2018); Kim et al. (2020) and score matching objectives Hyvärinen & Dayan (2005). Specifically, under additive Gaussian noise, the SURE cost function can be reformulated as an implicit score matching objective, differing only by a scaling factor and a constant term Kim & Ye (2021). In the context of MRI, recent innovations have capitalized on the intrinsic properties of data within a self-supervised framework Moreno López et al. (2021) to enhance denoising performance without clean labels. For standard MRI, Pfaff et al. (2023) utilizes SURE Kim et al. (2020) and spatially resolved noise maps to improve denoising performance, while Coil2Coil Park et al. (2022) leverages coil sensitivity to exploit multi-coil information effectively. In the realm of diffusion MRI (dMRI), specialized approaches have emerged to address the unique challenges of 4D data. Patch2Self Fadnavis et al. (2020) employs a J-invariant approach that preserves critical anatomical details by selectively excluding target volume data from its training inputs, though it requires a minimum of ten additional diffusion volumes. DDM2 Xiang et al. (2023) introduces a three-stage framework incorporating statistical image denoising into diffusion models. More recently, Wu et al. Wu et al. (2025) proposed Di-Fusion, a single-stage self-supervised approach that performs dMRI denoising by integrating statistical techniques with diffusion models through a Fusion process that prevents drift in results and a "Di-" process that better characterizes real-world noise distributions.

### 2.2 DENOISING SCORE MATCHING

Denoising Score Matching (DSM) Vincent (2011) is a framework for learning the score function (i.e., the gradient of the log-density) of the data distribution by training a model to reverse the corruption process of adding Gaussian noise. In DSM, given a clean data sample $\mathbf{X}_0 \in \mathbb{R}^d$ and a noise level $\sigma_t > 0$, a noisy observation $\mathbf{X}_t$ is generated by adding Gaussian noise: $\mathbf{X}_t = \mathbf{X}_0 + \sigma_t \mathbf{Z}, \quad \mathbf{Z} \sim \mathcal{N}(\mathbf{0}, \mathbf{I}_d)$. The objective is to train a denoising function $\mathbf{h}_\theta : \mathbb{R}^d \times \mathbb{R} \to \mathbb{R}^d$, parameterized by $\theta$, that estimates the clean image $\mathbf{X}_0$ from its noisy counterpart $\mathbf{X}_t$. This is achieved by minimizing the expected mean squared error (MSE)

loss $\mathbb{E}_{\mathbf{X}_0, \mathbf{X}_t} \left[ \|\mathbf{h}_\theta(\mathbf{X}_t, t) - \mathbf{X}_0\|^2 \right]$. Vincent (2011) demonstrated that optimizing the denoising function $\mathbf{h}_\theta$ using the MSE loss is equivalent to learning the score function of the noisy data distribution $p_t(\mathbf{x}_t)$ up to a scaling factor. Specifically, using *Tweedie's formula*, the relationship between the denoising function and the score function is given by: $\nabla_{\mathbf{x}_t} \log p_t(\mathbf{x}_t) = \frac{1}{\sigma_t^2} (\mathbf{h}_\theta(\mathbf{x}_t, t) - \mathbf{x}_t)$. By learning $\mathbf{h}_\theta$, we implicitly learn $\nabla_{\mathbf{x}_t} \log p_t(\mathbf{x}_t)$, which forms the foundation of score-based generative models Song & Ermon (2019); Ho et al. (2020). However, DSM relies on access to clean data during training, limiting its applicability in situations where only noisy observations are available. *Ambient Denoising Score Matching* (ADSM) Daras et al. (2024) leverages a double application of Tweedie's formula to relate noisy and clean distributions, enabling score-based learning from noisy observations. While ADSM was originally designed to mitigate memorization in large diffusion models by treating noisy data as a form of regularization Daras et al. (2024), its potential for self-supervised denoising remains underexplored. In our work, we bridge the gap of score-based self-supervised denoising in practical imaging applications.

## 3 METHODOLOGY

Consider a clean image $\mathbf{X}_0 \in \mathbb{R}^d$ and its corresponding noisy observation $\mathbf{X}_{t_{\text{data}}} \in \mathbb{R}^d$. We formulate the self-supervised denoising problem as estimating the conditional expectation $\mathbb{E}[\mathbf{X}_0 \mid \mathbf{X}_{t_{\text{data}}}]$, which constitutes the optimal estimator of $\mathbf{X}_0$ in the minimum mean square error (MMSE) sense. While estimating this typically requires clean or high-SNR reference images in a supervised learning setting, we adopt an ambient score-matching perspective inspired by Daras et al. (2024), circumventing the need for clean labels. The noisy image can be modeled as:

$$\mathbf{X}_{t_{\text{data}}} = \mathbf{X}_0 + \sigma_{t_{\text{data}}} \mathbf{N}, \tag{1}$$

where $\sigma_{t_{\text{data}}} > 0$ denotes the noise standard deviation at level $t_{\text{data}}$, and $\mathbf{N} \sim \mathcal{N}(\mathbf{0}, \mathbf{I}_d)$ represents the noise component. In MRI, the noise $\mathbf{N}$ is typically assumed to be additive, ergodic, stationary, uncorrelated, and white in k-space Liang & Lauterbur (2000). When the signal-to-noise ratio (SNR) exceeds two, the noise in the image domain can be well-approximated as Gaussian distributed Gudbjartsson & Patz (1995). While this Gaussian assumption underpins our theoretical framework, our empirical results suggest robustness even when this condition is not strictly satisfied.

The noise level $\sigma_{t_{\text{data}}}$ scales the noise to match the observed noise level in the MRI data. To facilitate self-supervised learning, we introduce a forward corruption process that systematically adds additional Gaussian noise to $\mathbf{X}_{t_{\text{data}}}$, defining a continuum of increasingly noisy versions of the data:

$$\mathbf{X}_t = \mathbf{X}_{t_{\text{data}}} + \sqrt{\sigma_t^2 - \sigma_{t_{\text{data}}}^2} \, \mathbf{Z}, \quad \mathbf{Z} \sim \mathcal{N}(\mathbf{0}, \mathbf{I}_d), \quad t > t_{\text{data}}, \tag{2}$$

where $\sigma_t$ is a strictly monotonically increasing noise schedule function for $t \in (t_{\text{data}}, T]$, with $T$ being the maximum noise level. This process allows us to model the distribution of the noisy data at different noise levels and forms the foundation for our generalized denoising score matching approach. In scenarios where the noise deviates from Gaussianity (e.g., Rician noise in low-SNR regions), pre-processing techniques such as the Variance Stabilizing Transform (VST) Foi (2011) can be applied to better approximate Gaussian statistics. Our goal is to train a denoising function $\mathbf{h}_\theta : \mathbb{R}^d \times \mathbb{R} \to \mathbb{R}^d$, parameterized by $\theta$, which maps a noisy input $\mathbf{X}_t$ at noise level $t$ to an estimate of either the clean image $\mathbf{X}_0$ or a less noisy version corresponding to a target noise level $\sigma_{t_{\text{target}}} \leq \sigma_{t_{\text{data}}}$, where $\mathbf{X}_{t_{\text{target}}} = \mathbf{X}_0 + \sigma_{t_{\text{target}}} \mathbf{N}_0$, with $\mathbf{N}_0 \sim \mathcal{N}(\mathbf{0}, \mathbf{I}_d)$.

### 3.1 GENERALIZED DENOISING SCORE MATCHING

We introduce the Generalized Denoising Score Matching (GDSM) loss, which enables learning a denoising function directly from noisy observations by modeling the conditional expectation of a higher-SNR image given a further corrupted version of the noisy data (proof provided in Appendix 4).

**Theorem 1** (Generalized Denoising Score Matching). *Let $\mathbf{X}_0 \in \mathbb{R}^d$ be a clean data sample drawn from the distribution $p_0(\mathbf{x}_0)$. Suppose that the noisy observation at a given data noise level $t_{data}$ is*

$$\mathbf{X}_{t_{data}} = \mathbf{X}_0 + \sigma_{t_{data}}\mathbf{N}, \quad \mathbf{N} \sim \mathcal{N}(\mathbf{0}, \mathbf{I}_d),$$

*and that for any $t > t_{data}$ the observation $\mathbf{X}_t$ is generated according to the forward process described in Equation equation 2. Let $\mathbf{h}_\theta : \mathbb{R}^d \times (t_{data}, T] \to \mathbb{R}^d$ be a denoising function parameterized by $\theta$, and fix a target noise level $\sigma_{t_{target}}$ satisfying $0 \le \sigma_{t_{target}} \le \sigma_{t_{data}}$. Define the loss function*

$$J(\theta) = \mathbb{E}_{\mathbf{X}_{t_{data}}, t, \mathbf{x}_t}\left[\left\|\gamma(t, \sigma_{t_{target}})\,\mathbf{h}_\theta(\mathbf{X}_t, t) + \delta(t, \sigma_{t_{target}})\,\mathbf{X}_t - \mathbf{X}_{t_{data}}\right\|^2\right], \tag{3}$$

*where $t$ is sampled uniformly from $(t_{data}, T]$ and the coefficients are defined by*

$$\gamma(t, \sigma_{t_{target}}) := \frac{\sigma_t^2 - \sigma_{t_{data}}^2}{\sigma_t^2 - \sigma_{t_{target}}^2} \quad and \quad \delta(t, \sigma_{t_{target}}) := \frac{\sigma_{t_{data}}^2 - \sigma_{t_{target}}^2}{\sigma_t^2 - \sigma_{t_{target}}^2}.$$

*Then any minimizer $\theta^*$ of $J(\theta)$ satisfies*

$$\mathbf{h}_{\theta^*}(\mathbf{X}_t, t) = \mathbb{E}\left[\mathbf{X}_{t_{target}} \mid \mathbf{X}_t\right].$$

*In other words, the optimal denoising function recovers the conditional expectation of the image with noise level $\sigma_{t_{target}}$ given the more heavily corrupted observation $\mathbf{X}_t$.*

**Remark 1.** *The proposed GDSM framework generalizes several existing methods: when $\sigma_{t_{target}} = \sigma_{t_{data}}$, GDSM reduces to the standard denoising score matching (DSM) Vincent (2011); when $\sigma_{t_{target}} = 0$, it recovers the ambient denoising score matching (ADSM) formulation Daras et al. (2024). Moreover, GDSM subsumes Noisier2Noise Moran et al. (2020) as a special case. By setting $\sigma_{t_{target}} = 0$ and fixing the noise level to $\sigma_t^2 = (1 + \alpha^2)\sigma_{t_{data}}^2$, the coefficients simplify to $\gamma(t, 0) = \frac{\alpha^2}{1+\alpha^2}$ and $\delta(t, 0) = \frac{1}{1+\alpha^2}$, thereby generalizing Noisier2Noise to a continuous range of noise levels. For further details, see Section B.2.*

To enhance training stability and improve convergence, we introduce a **reparameterization of the noise levels**. Let $\tau \in (0, T']$ be a new variable defined by

$$\sigma_\tau^2 = \sigma_t^2 - \sigma_{t_{\text{data}}}^2, \quad T' = \sqrt{\sigma_T^2 - \sigma_{t_{\text{data}}}^2}. \tag{4}$$

The original $t$ can be recovered via the inverse of $\sigma_t$, as:

$$t = \sigma_t^{-1}\left(\sqrt{\sigma_\tau^2 + \sigma_{t_{\text{data}}}^2}\right). \tag{5}$$

Since $\sigma_t$ is strictly increasing, $\sigma_t^{-1}$ is well-defined. In practice, as $T' \gg t_{\text{data}}$, we approximate $T' \approx T$ to allow uniform sampling over $\tau$ and consistent coverage of the noise level range during training, leading to smoother and faster convergence. To further improve training stability, we combine our reparameterization strategy with Exponential Moving Average (EMA) of model parameters. Details of the training dynamics with and without reparameterization (and the effect of EMA) are provided in Appendix I (see Figure 12).

Under this reparameterization, the loss function in Equation equation 12 becomes:

$$J'(\theta) = \mathbb{E}_{\mathbf{X}_{t_{\text{data}}}, \tau, \mathbf{X}_t}\left[\left\|\gamma'(\tau, \sigma_{t_{\text{target}}})\,\mathbf{h}_\theta(\mathbf{X}_t, t) + \delta'(\tau, \sigma_{t_{\text{target}}})\,\mathbf{X}_t - \mathbf{X}_{t_{\text{data}}}\right\|^2\right], \tag{6}$$

where the coefficients are:

$$\gamma'(\tau, \sigma_{t_{\text{target}}}) = \frac{\sigma_\tau^2}{\sigma_\tau^2 + \sigma_{t_{\text{data}}}^2 - \sigma_{t_{\text{target}}}^2}, \quad \delta'(\tau, \sigma_{t_{\text{target}}}) = \frac{\sigma_{t_{\text{data}}}^2 - \sigma_{t_{\text{target}}}^2}{\sigma_\tau^2 + \sigma_{t_{\text{data}}}^2 - \sigma_{t_{\text{target}}}^2}. \tag{7}$$

**Corollary 2** (Reparameterized Generalized Denoising Score Matching). *With our reparameterization strategy, the minimizer $\theta^*$ of the objective $J'(\theta)$ satisfies (proof provided in Appendix 5):*

$$\mathbf{h}_{\theta^*}(\mathbf{X}_t, t) = \mathbb{E}\left[\mathbf{X}_{t_{target}} \mid \mathbf{X}_t\right], \quad \forall \mathbf{X}_t \in \mathbb{R}^d, \, t > t_{data}.$$

## 3.2 CORRUPTION2SELF: A SCORE-BASED SELF-SUPERVISED MRI DENOISING FRAMEWORK

Building upon the Reparameterized Generalized Denoising Score Matching introduced earlier, we propose Corruption2Self (C2S) where the objective is to train a denoising function $\mathbf{h}_\theta$ that approximates the conditional expectation $\mathbb{E}\left[\mathbf{X}_{t_{\text{target}}} \mid \mathbf{X}_t\right]$, where $\mathbf{X}_{t_{\text{target}}}$ is a less noisy version of $\mathbf{X}_{t_{\text{data}}}$ with noise level $\sigma_{t_{\text{target}}} \leq \sigma_{t_{\text{data}}}$. Denote $\mathbf{D}_\theta(\mathbf{X}_\tau, \tau)$ as the weighted combination of the network output with the input through skip connections:

$$\mathbf{D}_\theta(\mathbf{X}_\tau, \tau) = \lambda_{\text{out}}(\tau, \sigma_{t_{\text{target}}})\, \mathbf{h}_\theta(\mathbf{X}_t, t) + \lambda_{\text{skip}}(\tau, \sigma_{t_{\text{target}}})\, \mathbf{X}_t, \qquad (8)$$

where $\mathbf{X}_t = \mathbf{X}_{t_{\text{data}}} + \sigma_\tau \mathbf{Z}$ with $\mathbf{Z} \sim \mathcal{N}(\mathbf{0}, \mathbf{I}_d)$, and $t$ relates to $\tau$ through $t = \sigma_t^{-1}\left(\sqrt{\sigma_\tau^2 + \sigma_{t_{\text{data}}}^2}\right)$. The blending coefficients are defined as:

$$\lambda_{\text{out}}(\tau, \sigma_{t_{\text{target}}}) = \frac{\sigma_\tau^2}{\sigma_\tau^2 + \sigma_{t_{\text{data}}}^2 - \sigma_{t_{\text{target}}}^2}, \quad \lambda_{\text{skip}}(\tau, \sigma_{t_{\text{target}}}) = \frac{\sigma_{t_{\text{data}}}^2 - \sigma_{t_{\text{target}}}^2}{\sigma_\tau^2 + \sigma_{t_{\text{data}}}^2 - \sigma_{t_{\text{target}}}^2}. \qquad (9)$$

When $\sigma_{t_{\text{target}}} = 0$ (maximum denoising), the goal is to predict $\mathbb{E}[\mathbf{X}_0 \mid \mathbf{X}_t]$ and the coefficients simplify to: $\lambda_{\text{out}}(\tau, 0) = \sigma_\tau^2/(\sigma_\tau^2 + \sigma_{t_{\text{data}}}^2), \quad \lambda_{\text{skip}}(\tau, 0) = \sigma_{t_{\text{data}}}^2/(\sigma_\tau^2 + \sigma_{t_{\text{data}}}^2))$. Our loss function is expressed as:

$$\mathcal{L}_{\text{C2S}}(\theta) = \frac{1}{2}\mathbb{E}_{\mathbf{X}_{t_{\text{data}}} \sim p_{t_{\text{data}}}(\mathbf{x}), \tau \sim \mathcal{U}(0, T]}\left[w(\tau)\left\|\mathbf{D}_\theta(\mathbf{X}_\tau, \tau) - \mathbf{X}_{t_{\text{data}}}\right\|_2^2\right], \qquad (10)$$

where $\mathbf{X}_\tau$ is constructed by adding noise to $\mathbf{X}_{t_{\text{data}}}$ according to the reparameterized noise level $\tau$, and $w(\tau)$ is a weighting function designed to balance the contributions from different noise levels. Following practices from prior works Song et al. (2020); Kingma et al. (2021); Karras et al. (2022), $w(\tau)$ can be set to $\left(\sigma_\tau^2 + \sigma_{t_{\text{data}}}^2\right)^\alpha$, with $\alpha$ being a hyperparameter controlling the weighting.

During inference, given a noisy observation $\mathbf{X}_{t_{\text{data}}}$, the denoised output is obtained by:

$$\hat{\mathbf{X}} = \mathbf{h}_{\theta^*}(\mathbf{X}_{t_{\text{data}}}, t_{\text{data}}), \qquad (11)$$

where the trained model $\mathbf{h}_{\theta^*}$ approximates $\mathbb{E}[\mathbf{X}_0 \mid \mathbf{X}_{t_{\text{data}}}]$, providing a clean estimate of the image.

While the C2S training procedure effectively approximates $\mathbb{E}[\mathbf{X}_0 \mid \mathbf{X}_t]$ when $\sigma_{t_{\text{target}}} = 0$, it can lead to oversmoothing. To maintain a balance between noise reduction and feature preservation, we introduce a **detail refinement** extension where the network is trained to predict $\mathbb{E}[\mathbf{X}_{t_{\text{target}}} \mid \mathbf{X}_t]$ with a non-zero target noise level $\sigma_{t_{\text{target}}} > 0$, allowing the network to retain a controlled amount of noise for preserving finer image textures (details are provided in Appendix G). As shown in Table 1, incorporating the detail refinement extension leads to a statistically significant improvement in image quality across contrasts. Our denoising model builds upon the U-Net architecture employed in DDPM Ho et al. (2020), enhanced with time conditioning and the Noise Variance Conditioned Multi-Head Self-Attention (NVC-MSA) module Hatamizadeh et al. (2023), which enables the self-attention layers to dynamically adapt to varying noise scales. Empirically, we found that setting the noise schedule function $\sigma_\tau$ equal to the noise level $\tau$ yields good performance. More details regarding the architecture and implementation are provided in Appendix C.

## 4 RESULTS AND DISCUSSION

We first evaluate C2S on the **M4Raw** dataset, which includes in-vivo MRI data with real noise. The training and validation data use three-repetition-averaged images, while the test data comprises higher-SNR labels,

**Algorithm 1** Corruption2Self Training Procedure

**Require:** Noisy dataset $X_{t_{data}}{}_{i=1}^{N}$, $h_\theta$, max noise level $T$, batch size $m$, total iterations $K$, noise schedule function $\sigma_\tau$
1: **for** $k = 1$ to $K$ **do**
2:     Sample minibatch $X_{t_{data}}{}_{i=1}^{m}$, $\tau \sim \mathcal{U}(0, T)$, $Z \sim \mathcal{N}(0, I_d)$
3:     Compute $\lambda_{out}(\tau, 0) = \frac{\sigma_\tau^2}{\sigma_\tau^2 + \sigma_{t_{data}}^2}$, $\lambda_{skip}(\tau, 0) = \frac{\sigma_{t_{data}}^2}{\sigma_\tau^2 + \sigma_{t_{data}}^2}$
4:     Recover $t = \sigma_t^{-1}(\sqrt{\sigma_\tau^2 + \sigma_{t_{data}}^2})$
5:     Compute $X_t \leftarrow X_{t_{data}} + \sigma_\tau Z$
6:     Compute loss: $\mathcal{L}_{C2S}(\theta) = \frac{1}{2m} \sum_{i=1}^{m} w(\tau) \| \cdot \|$
7:     Update $\theta$ using Adam optimizer Kingma (2014)

Table 1: Effectiveness of detail refinement module on the M4Raw validation dataset. Improvements are statistically significant ($*p < 0.05$) using paired t-tests. Additional details are provided in Appendix G.

| Contrast | PSNR | p-value |
|---|---|---|
| T1 | $34.56_{\pm 0.062} \rightarrow \mathbf{34.89}_{\pm 0.038}$ | $0.001^*$ |
| T2 | $33.84_{\pm 0.090} \rightarrow \mathbf{34.07}_{\pm 0.121}$ | $0.001^*$ |
| FLAIR | $32.44_{\pm 0.073} \rightarrow \mathbf{32.58}_{\pm 0.089}$ | $0.016^*$ |
| | **SSIM** | **p-value** |
| T1 | $0.885_{\pm 0.002} \rightarrow \mathbf{0.892}_{\pm 0.003}$ | $0.007^*$ |
| T2 | $0.860_{\pm 0.003} \rightarrow \mathbf{0.867}_{\pm 0.002}$ | $0.003^*$ |
| FLAIR | $0.812_{\pm 0.004} \rightarrow \mathbf{0.818}_{\pm 0.001}$ | $0.005^*$ |

created by averaging six repetitions for T1 and T2, and four for FLAIR. This setup allows us to assess how well denoising methods perform when evaluated on cleaner test data. Table 2 compares the performance of C2S against classical methods (NLM, BM3D), supervised learning (SwinIR, Restormer, Noise2Noise), and self-supervised approaches (Noise2Void, Noise2Self, PUCA, LG-BPN, Noisier2Noise, Recorrupted2Recorrupted). C2S consistently outperforms other self-supervised methods, achieving the highest PSNR and SSIM across all contrasts. Our base C2S model significantly outperforms existing self-supervised methods across all contrasts, achieving PSNRs of 32.59dB, 32.28dB, and 32.43dB for T1, T2, and FLAIR respectively. With our detail refinement extension, C2S further improves performance to 32.77dB/0.919, 32.33dB/0.890, and 32.51dB/0.876 for T1, T2, and FLAIR contrasts respectively. Notably, recent self-supervised methods like PUCA and LG-BPN, demonstrate lower performance on MRI data. This performance gap can be attributed to the blind-spot architecture design, often leading to information loss and oversmoothed results.

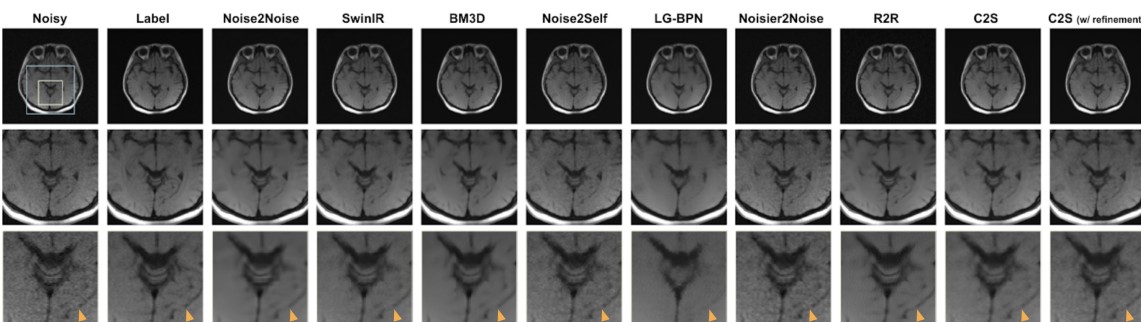

Figure 2: Comparison of different denoising methods for T1 contrast from the M4Raw dataset.

An important observation is that supervised methods such as SwinIR and Restormer, trained on three-repetition-averaged labels, do not significantly outperform self-supervised methods on higher-SNR test data. Supervised models typically learn $\mathbb{E}[\mathbf{X}_{t_{\text{target}}} \mid \mathbf{X}_{t_{\text{data}}}]$, where $t_{\text{data}} > t_{\text{target}} > 0$ when multi-repetition averaged samples are used as labels. This makes supervised methods less effective at handling shifts to higher-SNR test data. In contrast, C2S approximates $\mathbb{E}[\mathbf{X}_0 \mid \mathbf{X}_t]$, allowing it to achieve competitive performance on test data. Empirical results on test labels (three-repetition-average) matching the SNR of the training data (presented

| Methods | T1
PSNR / SSIM ↑ | T2
PSNR / SSIM ↑ | FLAIR
PSNR / SSIM ↑ |
|---|---|---|---|
| *Classical Non-Learning-Based Methods* | | | |
| NLM Froment (2014) | 31.90 / 0.898 | 31.17 / 0.876 | 32.01 / 0.870 |
| BM3D Mäkinen et al. (2020) | 32.07 / 0.903 | 31.20 / 0.877 | 32.14 / 0.873 |
| *Supervised Learning Methods* | | | |
| SwinIR Liang et al. (2021) | 32.53 / 0.913 | 31.90 / 0.891 | 32.15 / 0.885 |
| Restormer Zamir et al. (2022) | 32.35 / 0.912 | 31.79 / 0.890 | 32.31 / 0.886 |
| Noise2Noise Lehtinen et al. (2018) | 32.59 / 0.911 | 32.37 / 0.886 | 32.70 / 0.871 |
| *Self-Supervised Single-Contrast Methods* | | | |
| Noise2Void Krull et al. (2019) | 31.46 / 0.870 | 30.93 / 0.857 | 31.17 / 0.851 |
| Noise2Self Batson & Royer (2019) | 31.72 / 0.887 | 31.18 / 0.873 | 31.72 / 0.870 |
| PUCA Jang et al. (2024) | 30.52 / 0.870 | 29.11 / 0.827 | 29.57 / 0.807 |
| LG-BPN Wang et al. (2023) | 31.15 / 0.890 | 30.66 / 0.868 | 30.82 / 0.862 |
| Noisier2Noise Moran et al. (2020) | 31.60 / 0.876 | 31.45 / 0.871 | 31.59 / 0.861 |
| Recorrupted2Recorrupted Pang et al. (2021) | 31.67 / 0.876 | 31.33 / 0.870 | 31.57 / 0.863 |
| **C2S** | 32.59 / 0.915 | 32.28 / 0.888 | 32.43 / 0.872 |
| **C2S** *(w/ Detail Refinement)* | **32.77 / 0.919** | **32.33 / 0.890** | **32.51 / 0.876** |

Table 2: Quantitative comparison of denoising performance on the M4Raw test dataset. Results show PSNR (dB) and SSIM metrics across T1, T2, and FLAIR contrasts. C2S with detail refinement (bold) achieves the best performance among all self-supervised methods across all contrasts, and even outperforms supervised approaches in some cases. Second-best results among self-supervised methods are underlined.

in Appendix F) show that supervised methods like SwinIR and Restormer perform better when the noise characteristics of the training and test data are similar.

| Methods | **PD**, $\sigma = 13/255$
PSNR / SSIM ↑ | **PD**, $\sigma = 25/255$
PSNR / SSIM ↑ | **PDFS**, $\sigma = 13/255$
PSNR / SSIM ↑ | **PDFS**, $\sigma = 25/255$
PSNR / SSIM ↑ |
|---|---|---|---|---|
| *Classical Non-Learning-Based Methods* | | | | |
| NLM | 30.40 / 0.772 | 21.63 / 0.327 | 28.82 / 0.726 | 21.12 / 0.350 |
| BM3D | 33.16 / 0.829 | 30.58 / 0.755 | 30.64 / 0.705 | 28.49 / 0.592 |
| *Supervised Learning Methods* | | | | |
| Noise2True (SwinIR) | 34.44 / 0.868 | 32.39 / 0.820 | 31.35 / 0.774 | 29.55 / 0.665 |
| Noise2True (U-Net) | 34.54 / 0.870 | 32.61 / 0.825 | 31.39 / 0.775 | 29.62 / 0.669 |
| Noise2Noise Lehtinen et al. (2018) | 34.06 / 0.854 | 30.83 / 0.769 | 31.33 / 0.773 | 29.12 / 0.654 |
| *Self-Supervised Methods* | | | | |
| Noise2Void Krull et al. (2019) | 32.19 / 0.804 | 29.79 / 0.706 | 29.50 / 0.629 | 27.99 / 0.558 |
| Noise2Self Batson & Royer (2019) | 32.47 / 0.808 | 30.60 / 0.757 | 29.32 / 0.613 | 28.31 / 0.563 |
| PUCA Jang et al. (2024) | 31.03 / 0.771 | 29.88 / 0.740 | 28.65 / 0.594 | 27.54 / 0.527 |
| LG-BPN Wang et al. (2023) | 31.15 / 0.776 | 30.32 / 0.751 | 29.14 / 0.603 | 27.77 / 0.535 |
| Noisier2Noise Moran et al. (2020) | 33.18 / 0.807 | 30.35 / 0.741 | 30.39 / 0.683 | 27.83 / 0.559 |
| Recorrupted2Recorrupted Pang et al. (2021) | 33.29 / 0.810 | 30.52 / 0.745 | **30.95** / 0.752 | 28.32 / 0.561 |
| **C2S** | 33.36 / 0.831 | 30.62 / 0.753 | 30.72 / 0.750 | 28.58 / 0.592 |
| **C2S** *(w/ Detail Refinement)* | **33.48 / 0.832** | **30.67 / 0.761** | 30.91 / **0.756** | **28.62 / 0.601** |

Table 3: Quantitative evaluation on the fastMRI test dataset with simulated noise at two different levels ($\sigma = 13/255$ and $\sigma = 25/255$) across PD and PDFS contrasts.

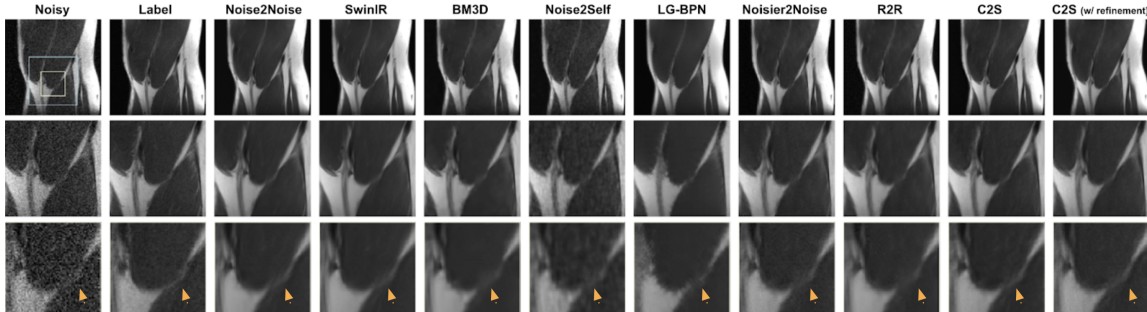

Figure 3: Comparison of denoising methods for the PD contrast ($\sigma = 13/255$) from the fastMRI dataset.

To further evaluate the robustness of C2S under different noise levels, we conducted experiments on the **fastMRI** dataset Zbontar et al. (2018), simulating Gaussian noise with $\sigma = 13/255$ and $\sigma = 25/255$. As shown in Table 3, the same baseline methods are analyzed and C2S consistently achieves the best or comparable results among self-supervised methods. On PDFS with $\sigma = 13/255$, Recorrupted2Recorrupted achieves a slightly higher PSNR (30.95 dB vs. 30.91 dB); however, C2S records the highest SSIM (0.756), indicating better detail preservation. It is worth noting that although the labels in this simulated dataset do not have added synthetic noise, they still contain inherent noise typical in MRI, albeit with higher SNR. Figure 3 demonstrates that our method balances feature preservation and noise removal, resulting in much cleaner visual representations compared to other methods. For additional results on fastMRI, refer to Appendix E.

We assessed the effect of reparameterization on training stability and performance by mapping the noise levels $\tau \in (0, T]$ to a new scale for more uniform sampling. As shown in Table 4a, reparameterization improves PSNR and SSIM across all contrasts. The training dynamics, illustrated in Figure 12, confirm the stabilizing effect of reparameterization. The model with reparameterization (blue) shows smoother and faster convergence than the model without it (orange), which fluctuates more and converges slower.

| Method | T1 | | T2 | | FLAIR | |
|---|---|---|---|---|---|---|
| | PSNR ↑ | SSIM ↑ | PSNR ↑ | SSIM ↑ | PSNR ↑ | SSIM ↑ |
| Without Reparam. | 31.14 | 0.837 | 30.53 | 0.807 | 30.43 | 0.771 |
| With Reparam. | **34.43** | **0.882** | **33.82** | **0.860** | **32.56** | **0.814** |

| Architecture | M4Raw | | fastMRI | |
|---|---|---|---|---|
| | PSNR ↑ | SSIM ↑ | PSNR ↑ | SSIM ↑ |
| U-Net | 33.11 | 0.865 | 32.32 | 0.807 |
| DDPM | 34.82 | 0.886 | 33.48 | 0.835 |
| **Ours** | **34.91** | **0.890** | **33.63** | **0.837** |

(a) Impact of reparameterization of noise levels on the M4Raw dataset. Results are validation results obtained after training for 200 epochs.

(b) Influence of model architecture on the M4Raw dataset (T1) and fastMRI dataset (PD).

Table 4: Ablation studies on the impact of reparameterization and model architecture.

We evaluated the impact of different architectural choices on the performance of our denoising model. As shown in Table 4b, incorporating time conditioning significantly improves both PSNR and SSIM across the M4Raw (T1 contrast) and fastMRI (PD contrast, noise level 13/255) datasets. The best performance is achieved by further incorporating the NVC-MSA module in our model (Appendix C), which allows the model to dynamically adapt to varying noise levels by integrating noise variance into the self-attention mechanism.

Our approach demonstrates strong robustness to noise level estimation errors, making it suitable for practical applications where exact noise levels may be unknown. Through extensive experiments detailed in Appendix H, we show that C2S maintains stable performance even with significant estimation errors (±50% of the true noise level). This robustness, combined with the incorporation of standard noise estimation

techniques (e.g., from the `skimage` package Van der Walt et al. (2014)), enables our method to effectively function as a blind denoising model. In practice, we find such noise estimation tools provide sufficiently accurate estimates for optimal model performance, alleviating the need for precise noise level knowledge. More quantitative results and analysis of the effect of noise estimation error are provided in Appendix H.

**Extending C2S to Multi-Contrast Settings**   MRI typically involves acquiring multiple contrasts to provide comprehensive diagnostic information. By leveraging complementary information from different contrasts, denoising performance can be further enhanced. To capitalize on this, we extend the C2S framework to multi-contrast settings by incorporating additional MRI contrasts as inputs. Figure 4 demonstrates the visual comparison of different denoising methods for the T1 contrast on the M4Raw dataset. It is evident that using multi-contrast inputs (T1 & T2, T1 & FLAIR) allows for better structural preservation and more detailed reconstructions compared to single-contrast denoising techniques. Quantitatively, Table 5 shows that multi-contrast C2S consistently outperforms classical BM3D, supervised Noise2Noise, and single-contrast C2S in terms of PSNR and SSIM. More details can be found in Appendix D.

| Target Contrast | Best Classical | Best Supervised | Best Self-Supervised | Multi-Contrast C2S | | |
|---|---|---|---|---|---|---|
| PSNR / SSIM ↑ | BM3D | Noise2Noise | C2S | T1 & T2 | FLAIR & T1 | T2 & FLAIR |
| T1 | 32.07 / 0.903 | 32.59 / 0.911 | 32.77 / 0.919 | 33.57 / 0.921 | **33.89 / 0.922** | N/A |
| T2 | 31.20 / 0.877 | 32.37 / 0.886 | 32.33 / 0.890 | 33.01 / 0.895 | N/A | **33.36 / 0.901** |
| FLAIR | 32.14 / 0.873 | 32.70 / 0.871 | 32.51 / 0.876 | N/A | **32.71 / 0.879** | 32.62 / 0.877 |

Table 5: Multi-contrast denoising results on the M4Raw dataset. For the multi-contrast C2S results, entries such as "T1 & T2" indicate that T1 and T2 contrasts were used as inputs for denoising the target contrast.

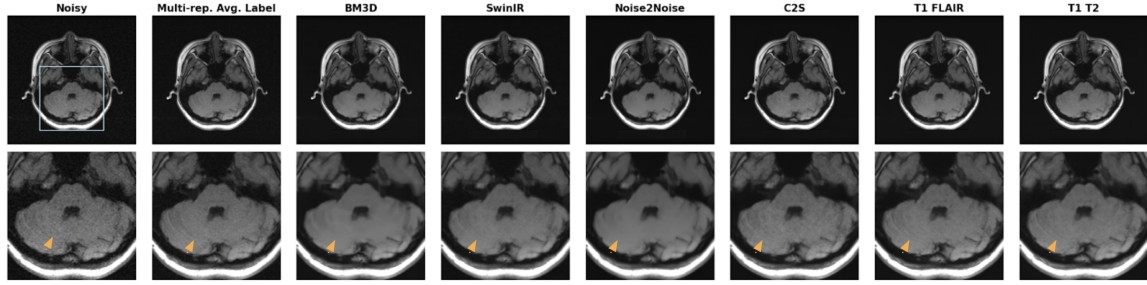

Figure 4: Comparison of different denoising methods for T1 contrast in the M4Raw dataset. The figure showcases Noisy, BM3D, Noise2Noise, and C2S, along with multi-contrast C2S variants (T1 & T2, T1 & FLAIR). Multi-contrast C2S preserves more structural details and produces sharper reconstructions.

## 5   CONCLUSION

We have introduced Corruption2Self, a score-based self-supervised denoising framework tailored for MRI applications. By extending denoising score matching to the ambient noise setting through our Generalized Denoising Score Matching approach, C2S enables effective learning directly from noisy observations without the need for clean labels. Our method incorporates a reparameterization of noise levels to stabilize training and enhance convergence, as well as a detail refinement extension to balance noise reduction with the preservation of fine spatial features. By extending C2S to multi-contrast settings, we further leverage complementary information across different MRI contrasts, leading to enhanced denoising performance. Notably, C2S exhibits superior robustness across varying noise conditions and MRI contrasts, highlighting its potential for broader applicability in clinical settings.

ACKNOWLEDGEMENTS

This research was partially supported by the following fundings: NSF-CBET-1944249 and NIH-R35GM142969. The authors would also like to thank Ruiyang Zhao and Yizun Wang for the helpful discussions.

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

# A    ADDITIONAL EXPERIMENTAL DETAILS

## A.1    DATASETS

We evaluated our Corruption2Self (C2S) method on two distinct datasets:

**In-vivo Dataset (M4Raw):** The M4Raw dataset Lyu et al. (2023) contains multi-channel k-space images across three contrasts (T1-weighted, T2-weighted, and FLAIR) from 183 participants. The dataset was partitioned into training (128 individuals, 6,912 slices), validation (30 individuals, 1,620 slices), and testing (25 individuals, 1,350 slices) subsets. For training and validation, pseudo-ground truth labels were generated by averaging three repetitions for T1/T2-weighted contrasts and two repetitions for FLAIR. Test data utilized higher-SNR labels derived from averaging six repetitions (T1/T2-weighted) and four repetitions (FLAIR), enabling assessment of model generalization to cleaner data. The dataset (20.7GB) is distributed under CC-BY-4.0 license.

**Simulated Dataset (fastMRI):** We utilized single-coil knee data from the fastMRI dataset Zbontar et al. (2018), selecting patient entries matching those in MINet Feng et al. (2021) to ensure contrast correspondence. The dataset was structured into training (180 pairs, 6,648 slices), validation (47 pairs, 1,684 slices), and testing (45 pairs, 1,665 slices) subsets. White Gaussian noise with $\sigma \in [13, 25]$ was introduced to simulate real-world MRI noise patterns. The dataset (31.5GB) is distributed under MIT license.

## A.2    TRAINING PARAMETERS

All experiments were conducted on NVIDIA A6000 GPUs. For the M4Raw dataset, optimization was performed using Adam with learning rate $1 \times 10^{-4}$ and weight decay $1 \times 10^{-4}$. For the fastMRI dataset, Adam was configured with learning rate $1 \times 10^{-4}$ and weight decay $5 \times 10^{-2}$. Critical hyperparameters (learning rate, weight decay, batch size, maximum noise level $T$) were optimized based on validation performance. Early stopping was implemented to prevent overfitting, and final models were selected based on optimal validation metrics before test set evaluation.

## A.3    EVALUATION PROTOCOL

To assess the efficacy and robustness of the Corruption2Self (C2S) framework, we employed the following evaluation protocol:

**Pseudo-Ground Truth Generation:** For the in-vivo dataset (M4Raw), pseudo-ground truth labels were generated by averaging the multi-repetition images.

**Image Quality Metrics:** The quality of the denoised images was quantified using two metrics:

- **Peak Signal-to-Noise Ratio (PSNR):** This metric measures the ratio between the maximum possible power of a signal and the power of corrupting noise, assessing the quality of the denoised image against the reference label.
- **Structural Similarity Index Measure (SSIM):** This metric evaluates the structural similarity between the denoised image and the reference, focusing on luminance, contrast, and structure.

# B    THEORETICAL RESULTS

**Lemma 3.** *Given the objective function $J(\theta)$:*

$$J(\theta) = \mathbb{E}_{\mathbf{X}_{t_{data}}} \left[ \mathbb{E}_t \left[ \mathbb{E}_{\mathbf{X}_t | \mathbf{X}_{t_{data}}} \left[ \|\mathbf{g}_\theta(\mathbf{X}_t, t) - \mathbf{X}_{t_{data}}\|^2 \right] \right] \right],$$

*where* $\mathbf{X}_t \sim \mathcal{N}(\mathbf{X}_{t_{data}}, \sigma^2(t)\mathbf{I})$, *the function* $\mathbf{g}_{\theta^*}(\mathbf{X}_t, t)$ *that minimizes* $J(\theta)$ *is the conditional expectation:*

$$\mathbf{g}_{\theta^*}(\mathbf{X}_t, t) = \mathbb{E}\left[\mathbf{X}_{t_{data}} \mid \mathbf{X}_t\right].$$

*Proof.* Our goal is to find $\mathbf{g}_{\theta^*}(\mathbf{X}_t, t)$ that minimizes the objective function $J(\theta)$. Since the outer expectations over $\mathbf{X}_{t_{\text{data}}}$ and $t$ do not affect the minimization with respect to $\theta$, we focus on minimizing the inner expectation:

$$\mathbb{E}_{\mathbf{X}_t | \mathbf{X}_{t_{\text{data}}}}\left[\left\|\mathbf{g}_\theta(\mathbf{X}_t, t) - \mathbf{X}_{t_{\text{data}}}\right\|^2\right].$$

For fixed $\mathbf{X}_t$ and $t$, the optimal function $\mathbf{g}_{\theta^*}(\mathbf{X}_t, t)$ minimizes the expected squared error:

$$\min_{\mathbf{g}_\theta} \mathbb{E}_{\mathbf{X}_{t_{\text{data}}} | \mathbf{X}_t}\left[\left\|\mathbf{g}_\theta(\mathbf{X}_t, t) - \mathbf{X}_{t_{\text{data}}}\right\|^2\right].$$

According to estimation theory, the function that minimizes this expected squared error is the conditional expectation of $\mathbf{X}_{t_{\text{data}}}$ given $\mathbf{X}_t$:

$$\mathbf{g}_{\theta^*}(\mathbf{X}_t, t) = \mathbb{E}\left[\mathbf{X}_{t_{\text{data}}} \mid \mathbf{X}_t\right].$$

Therefore, the function $\mathbf{g}_{\theta^*}(\mathbf{X}_t, t) = \mathbb{E}\left[\mathbf{X}_{t_{\text{data}}} \mid \mathbf{X}_t\right]$ minimizes the objective function $J(\theta)$. $\qquad\square$

**Theorem 4** (Generalized Denoising Score Matching; restated Theorem 1). *Let the following assumptions hold:*

1. **Data Distribution**: *The clean data vector* $\mathbf{X}_0 \in \mathbb{R}^d$ *is distributed according to* $p_0(\mathbf{x}_0)$.

2. **Noise Level Functions**: *The noise level or schedule functions* $\sigma_t$ *are strictly positive, scalar-valued functions of time* $t$, *and are monotonically increasing with respect to* $t$.

3. **Noisy Observations at Data Noise Level** $t_{\textbf{\textit{data}}}$: *The observed data* $\mathbf{X}_{t_{data}}$ *is given by* $\mathbf{X}_{t_{data}} = \mathbf{X}_0 + \sigma_{t_{data}}\mathbf{Z}_{t_{data}}$, *where* $\mathbf{Z}_{t_{data}} \sim \mathcal{N}(\mathbf{0}, \mathbf{I}_d)$.

4. **Target Noise Level** $t_{\textbf{\textit{target}}} \leq t_{\textbf{\textit{data}}}$: *The target noisy data* $\mathbf{X}_{t_{target}}$ *is defined as* $\mathbf{X}_{t_{target}} = \mathbf{X}_0 + \sigma_{t_{target}}\mathbf{Z}_{t_{target}}$, *where* $\mathbf{Z}_{t_{target}} \sim \mathcal{N}(\mathbf{0}, \mathbf{I}_d)$.

5. **Higher Noise Levels** $t \geq t_{\textbf{\textit{data}}}$: *For any* $t \geq t_{data}$, *the noisy data* $\mathbf{X}_t$ *is given by* $\mathbf{X}_t = \mathbf{X}_0 + \sigma_t\mathbf{Z}_t$, *where* $\mathbf{Z}_t \sim \mathcal{N}(\mathbf{0}, \mathbf{I}_d)$.

6. **Function Class Expressiveness**: *The neural network class* $\{\mathbf{h}_\theta\}$ *is sufficiently expressive, satisfying the universal approximation property.*

*Define the objective function:*

$$J(\theta) = \mathbb{E}_{\mathbf{X}_{t_{data}}, t, \mathbf{X}_t}\left[\left\|\gamma(t, \sigma_{t_{target}})\,\mathbf{h}_\theta(\mathbf{X}_t, t) + \delta(t, \sigma_{t_{target}})\,\mathbf{X}_t - \mathbf{X}_{t_{data}}\right\|^2\right], \tag{12}$$

*where* $t$ *is uniformly sampled from* $(t_{data}, T]$ *and*

$$\gamma(t, \sigma_{t_{target}}) = \frac{\sigma_t^2 - \sigma_{t_{data}}^2}{\sigma_t^2 - \sigma_{t_{target}}^2}, \quad \delta(t, \sigma_{t_{target}}) = \frac{\sigma_{t_{data}}^2 - \sigma_{t_{target}}^2}{\sigma_t^2 - \sigma_{t_{target}}^2}.$$

*Given* $\mathbf{X}_{t_{data}}$ *and* $t \geq t_{data}$, $\mathbf{X}_t$ *is sampled as:*

$$\mathbf{X}_t = \mathbf{X}_{t_{data}} + \sqrt{\sigma_t^2 - \sigma_{t_{data}}^2}\,\mathbf{Z}, \quad where \quad \mathbf{Z} \sim \mathcal{N}(\mathbf{0}, \mathbf{I}_d). \tag{13}$$

*Then, the minimizer* $\theta^*$ *of* $J(\theta)$ *satisfies:*

$$\mathbf{h}_{\theta^*}(\mathbf{X}_t, t) = \mathbb{E}[\mathbf{X}_{t_{target}} \mid \mathbf{X}_t], \quad \forall \mathbf{X}_t \in \mathbb{R}^d, \ t \geq t_{data}. \tag{14}$$

*Proof.* Our goal is to find $\theta^*$ that minimizes $J(\theta)$. Note that $\mathbf{X}_t$ can be expressed in terms of both $\mathbf{X}_{t_{\text{data}}}$ and $\mathbf{X}_{t_{\text{target}}}$:

$$\mathbf{X}_t = \mathbf{X}_{t_{\text{data}}} + \sqrt{\sigma_t^2 - \sigma_{t_{\text{data}}}^2}\, \mathbf{Z}_1, \quad \mathbf{Z}_1 \sim \mathcal{N}(\mathbf{0}, \mathbf{I}_d),$$

$$\mathbf{X}_t = \mathbf{X}_{t_{\text{target}}} + \sqrt{\sigma_t^2 - \sigma_{t_{\text{target}}}^2}\, \mathbf{Z}_2, \quad \mathbf{Z}_2 \sim \mathcal{N}(\mathbf{0}, \mathbf{I}_d).$$

Using properties of Gaussian distributions, the score function $\nabla_{\mathbf{X}_t} \log p_t(\mathbf{X}_t)$ can be written in two ways:

$$\nabla_{\mathbf{X}_t} \log p_t(\mathbf{X}_t) = \frac{\mathbb{E}\left[\mathbf{X}_{t_{\text{data}}} \mid \mathbf{X}_t\right] - \mathbf{X}_t}{\sigma_t^2 - \sigma_{t_{\text{data}}}^2} = \frac{\mathbb{E}\left[\mathbf{X}_{t_{\text{target}}} \mid \mathbf{X}_t\right] - \mathbf{X}_t}{\sigma_t^2 - \sigma_{t_{\text{target}}}^2}.$$

Equating these expressions and rearranging terms:

$$\mathbb{E}\left[\mathbf{X}_{t_{\text{data}}} \mid \mathbf{X}_t\right] = \frac{\sigma_t^2 - \sigma_{t_{\text{data}}}^2}{\sigma_t^2 - \sigma_{t_{\text{target}}}^2} \mathbb{E}\left[\mathbf{X}_{t_{\text{target}}} \mid \mathbf{X}_t\right] + \frac{\sigma_{t_{\text{data}}}^2 - \sigma_{t_{\text{target}}}^2}{\sigma_t^2 - \sigma_{t_{\text{target}}}^2} \mathbf{X}_t.$$

Recognizing the coefficients as $\gamma(t, \sigma_{t_{\text{target}}})$ and $\delta(t, \sigma_{t_{\text{target}}})$, respectively:

$$\mathbb{E}\left[\mathbf{X}_{t_{\text{data}}} \mid \mathbf{X}_t\right] = \gamma(t, \sigma_{t_{\text{target}}})\mathbb{E}\left[\mathbf{X}_{t_{\text{target}}} \mid \mathbf{X}_t\right] + \delta(t, \sigma_{t_{\text{target}}})\mathbf{X}_t.$$

Define the auxiliary function:

$$\mathbf{g}_\theta(\mathbf{X}_t, t) = \gamma(t, \sigma_{t_{\text{target}}})\mathbf{h}_\theta(\mathbf{X}_t, t) + \delta(t, \sigma_{t_{\text{target}}})\mathbf{X}_t.$$

Substituting into the loss function $J(\theta)$, we have:

$$J(\theta) = \mathbb{E}\left[\left\|\mathbf{g}_\theta(\mathbf{X}_t, t) - \mathbf{X}_{t_{\text{data}}}\right\|^2\right].$$

This is a mean squared error (MSE) objective between $\mathbf{g}_\theta(\mathbf{X}_t, t)$ and $\mathbf{X}_{t_{\text{data}}}$.

The objective function becomes:

$$J(\theta) = \mathbb{E}_{\mathbf{X}_{t_{\text{data}}}} \mathbb{E}_t \mathbb{E}_{\mathbf{X}_t \mid \mathbf{X}_{t_{\text{data}}}}\left[\left\|\mathbf{g}_\theta(\mathbf{X}_t, t) - \mathbf{X}_{t_{\text{data}}}\right\|^2\right].$$

By the property of MSE minimization (Lemma 3), the function $\mathbf{g}_{\theta^*}(\mathbf{X}_t, t)$ that minimizes $J(\theta)$ satisfies:

$$\mathbf{g}_{\theta^*}(\mathbf{X}_t, t) = \mathbb{E}\left[\mathbf{X}_{t_{\text{data}}} \mid \mathbf{X}_t\right].$$

Substituting the earlier expression for $\mathbb{E}\left[\mathbf{X}_{t_{\text{data}}} \mid \mathbf{X}_t\right]$:

$$\gamma(t, \sigma_{t_{\text{target}}})\mathbf{h}_{\theta^*}(\mathbf{X}_t, t) + \delta(t, \sigma_{t_{\text{target}}})\mathbf{X}_t = \gamma(t, \sigma_{t_{\text{target}}})\mathbb{E}\left[\mathbf{X}_{t_{\text{target}}} \mid \mathbf{X}_t\right] + \delta(t, \sigma_{t_{\text{target}}})\mathbf{X}_t.$$

Subtracting $\delta(t, \sigma_{t_{\text{target}}})\mathbf{X}_t$ from both sides:

$$\gamma(t, \sigma_{t_{\text{target}}})\mathbf{h}_{\theta^*}(\mathbf{X}_t, t) = \gamma(t, \sigma_{t_{\text{target}}})\mathbb{E}\left[\mathbf{X}_{t_{\text{target}}} \mid \mathbf{X}_t\right].$$

Since $\gamma(t, \sigma_{t_{\text{target}}}) > 0$ (due to $\sigma_t$ being strictly increasing and $t \geq t_{\text{data}}$), we can divide both sides by $\gamma(t, \sigma_{t_{\text{target}}})$:

$$\mathbf{h}_{\theta^*}(\mathbf{X}_t, t) = \mathbb{E}\left[\mathbf{X}_{t_{\text{target}}} \mid \mathbf{X}_t\right].$$

This completes the proof. $\square$

**Corollary 5** (Reparameterized Generalized Denoising Score Matching; restated Corollary 2)**.** *Let the assumptions of the Generalized Denoising Score Matching theorem 4 hold. Additionally, define:*

1. **Reparameterization**: *For $t \geq t_{data}$, define $\tau$ such that*

$$\sigma_\tau^2 = \sigma_t^2 - \sigma_{t_{data}}^2, \tag{15}$$

*where $\tau \in (0, T']$ and $T'$ is determined by $\sigma_{T'}^2 = \sigma_T^2 - \sigma_{t_{data}}^2$. Note that given $\tau$ and $t_{data}$, we can recover $t$ using the inverse function of $\sigma_t$, denoted as $\sigma_t^{-1}$:*

$$t = \sigma_t^{-1}(\sqrt{\sigma_\tau^2 + \sigma_{t_{data}}^2}). \tag{16}$$

*This inverse function exists and is well-defined due to the strictly monotonically increasing property of $\sigma_t$.*

*Define the new objective function:*

$$J'(\theta) = \mathbb{E}_{\mathbf{X}_{t_{data}}, \tau, \mathbf{X}_t} \left[ \left\| \gamma'(\tau, \sigma_{t_{target}}) \mathbf{h}_\theta(\mathbf{X}_t, t) + \delta'(\tau, \sigma_{t_{target}}) \mathbf{X}_t - \mathbf{X}_{t_{data}} \right\|^2 \right], \tag{17}$$

*where $\tau$ is uniformly sampled from $[0, T']$, and the coefficients are defined as:*

$$\gamma'(\tau, \sigma_{t_{target}}) = \frac{\sigma_\tau^2}{\sigma_\tau^2 + \sigma_{t_{data}}^2 - \sigma_{t_{target}}^2}, \quad \delta'(\tau, \sigma_{t_{target}}) = \frac{\sigma_{t_{data}}^2 - \sigma_{t_{target}}^2}{\sigma_\tau^2 + \sigma_{t_{data}}^2 - \sigma_{t_{target}}^2}.$$

*Given $\mathbf{X}_{t_{data}}$ and $\tau$, $\mathbf{X}_t$ is sampled as:*

$$\mathbf{X}_t = \mathbf{X}_{t_{data}} + \sigma_\tau \mathbf{Z}', \quad where \quad \mathbf{Z}' \sim \mathcal{N}(\mathbf{0}, \mathbf{I}_d). \tag{18}$$

*Then, the minimizer $\theta^*$ of $J'(\theta)$ satisfies:*

$$\mathbf{h}_{\theta^*}(\mathbf{X}_t, t) = \mathbb{E}[\mathbf{X}_{t_{target}} \mid \mathbf{X}_t], \quad \forall \mathbf{X}_t \in \mathbb{R}^d, \ t \geq t_{data}. \tag{19}$$

*Proof.* Substituting $\sigma_t^2$ into $\gamma(t, \sigma_{t_{\text{target}}})$ and $\delta(t, \sigma_{t_{\text{target}}})$ from Theorem 1, we obtain:

$$\gamma'(\tau, \sigma_{t_{\text{target}}}) = \gamma(t, \sigma_{t_{\text{target}}}) = \frac{\sigma_\tau^2}{\sigma_\tau^2 + \sigma_{t_{\text{data}}}^2 - \sigma_{t_{\text{target}}}^2}, \quad \delta'(\tau, \sigma_{t_{\text{target}}}) = \delta(t, \sigma_{t_{\text{target}}}) = \frac{\sigma_{t_{\text{data}}}^2 - \sigma_{t_{\text{target}}}^2}{\sigma_\tau^2 + \sigma_{t_{\text{data}}}^2 - \sigma_{t_{\text{target}}}^2}.$$

Define $\mathbf{g}_\theta(\mathbf{X}_t, t) = \gamma'(\tau, \sigma_{t_{\text{target}}})\mathbf{h}_\theta(\mathbf{X}_t, t) + \delta'(\tau, \sigma_{t_{\text{target}}})\mathbf{X}_t$. The objective function becomes:

$$J'(\theta) = \mathbb{E}_{\mathbf{X}_{t_{\text{data}}}} \mathbb{E}_\tau \mathbb{E}_{\mathbf{X}_t | \mathbf{X}_{t_{\text{data}}}} \left[ \left\| \mathbf{g}_\theta(\mathbf{X}_t, t) - \mathbf{X}_{t_{\text{data}}} \right\|^2 \right].$$

By Lemma 3, the function minimizing $J'(\theta)$ is:

$$\mathbf{g}_{\theta^*}(\mathbf{X}_t, t) = \mathbb{E}\left[\mathbf{X}_{t_{\text{data}}} \mid \mathbf{X}_t\right].$$

From proof of Theorem 1, we have:

$$\mathbb{E}\left[\mathbf{X}_{t_{\text{data}}} \mid \mathbf{X}_t\right] = \gamma'(\tau, \sigma_{t_{\text{target}}})\mathbb{E}\left[\mathbf{X}_{t_{\text{target}}} \mid \mathbf{X}_t\right] + \delta'(\tau, \sigma_{t_{\text{target}}})\mathbf{X}_t.$$

Therefore,

$$\mathbf{g}_{\theta^*}(\mathbf{X}_t, t) = \gamma'(\tau, \sigma_{t_{\text{target}}})\mathbf{h}_{\theta^*}(\mathbf{X}_t, t) + \delta'(\tau, \sigma_{t_{\text{target}}})\mathbf{X}_t = \gamma'(\tau, \sigma_{t_{\text{target}}})\mathbb{E}\left[\mathbf{X}_{t_{\text{target}}} \mid \mathbf{X}_t\right] + \delta'(\tau, \sigma_{t_{\text{target}}})\mathbf{X}_t.$$

Comparing both expressions, we conclude:

$$\gamma'(\tau, \sigma_{t_{\text{target}}})\mathbf{h}_{\theta^*}(\mathbf{X}_t, t) = \gamma'(\tau, \sigma_{t_{\text{target}}})\mathbb{E}\left[\mathbf{X}_{t_{\text{target}}} \mid \mathbf{X}_t\right].$$

Since $\gamma'(\tau, \sigma_{t_{\text{target}}}) > 0$, we divide both sides by $\gamma'(\tau, \sigma_{t_{\text{target}}})$:

$$\mathbf{h}_{\theta^*}(\mathbf{X}_t, t) = \mathbb{E}\left[\mathbf{X}_{t_{\text{target}}} \mid \mathbf{X}_t\right].$$

This completes the proof. $\qquad\square$

## B.1 EXTENSION TO VARIANCE PRESERVING CASE

Our GDSM framework can be naturally extended to the variance preserving (VP) Song et al. (2020); Ho et al. (2020) case. For example, when $\sigma_{t_{\text{target}}} = 0$, which corresponds to the VP formulation in ADSM Daras et al. (2024). In this case, the data model follows:

$$\mathbf{X}_{t_{\text{data}}} = \sqrt{1 - \sigma_{t_{\text{data}}}^2}\mathbf{X}_0 + \sigma_{t_{\text{data}}}\mathbf{Z}, \quad 0 < \sigma_{t_{\text{data}}} < 1 \tag{20}$$

Let $\mathbf{X}_0 \in \mathbb{R}^d$ represent clean data, and for any $\sigma_{t_{\text{data}}} < \sigma_t < 1$, the forward corrupted $\mathbf{X}_t$ is given by:

$$\mathbf{X}_t = \sqrt{1 - \sigma_t^2}\mathbf{X}_0 + \sigma_t\mathbf{Z}_t, \quad \mathbf{Z}_t \sim \mathcal{N}(\mathbf{0}, \mathbf{I}_d) \tag{21}$$

Define the objective function:

$$L_{\text{VP}}(\theta) = \mathbb{E}_{\mathbf{X}_{t_{\text{data}}}, t, \mathbf{X}_t}\left[\left\|\frac{\sigma_t^2}{\sigma_t^2 - \sigma_{t_{\text{data}}}^2}\sqrt{1 - \sigma_{t_{\text{data}}}^2}\,\mathbf{h}_\theta(\mathbf{X}_t, t) - \sigma_{t_{\text{data}}}^2\frac{\sqrt{1 - \sigma_t^2}}{\sigma_t^2 - \sigma_{t_{\text{data}}}^2}\mathbf{X}_t - \mathbf{X}_{t_{\text{data}}}\right\|^2\right] \tag{22}$$

Then, the minimizer $\theta^*$ of $J_{\text{VP}}(\theta)$ satisfies:

$$\mathbf{h}_{\theta^*}(\mathbf{X}_t, t) = \mathbb{E}[\mathbf{X}_0 \mid \mathbf{X}_t], \quad \forall \mathbf{X}_t \in \mathbb{R}^d, \ t \geq t_{\text{data}}. \tag{23}$$

## B.2 CONNECTION TO NOISIER2NOISE

We demonstrate that Noisier2Noise Moran et al. (2020) emerges as a special case of our Generalized Denoising Score Matching (GDSM) framework under specific conditions. Let us establish the correspondence between notations: in Noisier2Noise, $X$ represents the clean image, $Y = X + N$ represents the noisy observation with noise $N$, and $Z = Y + M$ represents the doubly-noisy image with additional synthetic noise $M$. These correspond to our formulation where $X$ is $\mathbf{X}_0$, $Y$ is $\mathbf{X}_{t_{\text{data}}}$, and $Z$ is $\mathbf{X}_t$.

From proof of Theorem 1, when setting $\sigma_{t_{\text{target}}} = 0$, we obtain:

$$\mathbb{E}[\mathbf{X}_{t_{\text{data}}} \mid \mathbf{X}_t] = \frac{\sigma_t^2 - \sigma_{t_{\text{data}}}^2}{\sigma_t^2}\mathbb{E}[\mathbf{X}_0 \mid \mathbf{X}_t] + \frac{\sigma_{t_{\text{data}}}^2}{\sigma_t^2}\mathbf{X}_t \tag{24}$$

In the improved variant of Noisier2Noise, a parameter $\alpha$ controls the magnitude of synthetic noise $M$ relative to the original noise $N$. We can establish that this parameter corresponds to our noise schedule through:

$$\alpha^2 = \frac{\sigma_t^2 - \sigma_{t_{\text{data}}}^2}{\sigma_{t_{\text{data}}}^2} \tag{25}$$

Under this relationship, Equation equation 24 becomes equivalent to the Noisier2Noise formulation:

$$\mathbb{E}[Y|Z] = \frac{\alpha^2}{1 + \alpha^2}\mathbb{E}[X|Z] + \frac{1}{1 + \alpha^2}Z \tag{26}$$

This equivalence leads to the characteristic Noisier2Noise correction formula:

$$\mathbb{E}[X|Z] = \frac{(1 + \alpha^2)\mathbb{E}[Y|Z] - Z}{\alpha^2} \tag{27}$$

In the standard case where $\alpha = 1$, this reduces to $\mathbb{E}[X|Z] = 2\mathbb{E}[Y|Z] - Z$, which corresponds to our framework with $\sigma_t^2 = 2\sigma_{t_{\text{data}}}^2$.

Our GDSM framework offers several advances over Noisier2Noise. First, it provides a continuous noise schedule through $\sigma_t$, allowing the model to learn from a spectrum of noise levels rather than a fixed ratio determined by $\alpha$. Second, it introduces explicit time conditioning in the network architecture, enabling better adaptation to different noise magnitudes. Third, and perhaps most importantly, it eliminates the need to tune the $\alpha$ parameter, which according to Moran et al. (2020) is "difficult or impossible to derive in the absence of clean validation data." Instead, our approach automatically learns to handle different noise levels through the continuous schedule and time conditioning. Furthermore, GDSM extends beyond the clean image prediction task by supporting arbitrary target noise levels through $\sigma_{t_{\text{target}}}$, providing a unified framework for various denoising objectives.

## C   MODEL ARCHITECTURE

In this appendix, we provide a comprehensive description of the architectures employed for both single-contrast and multi-contrast MRI denoising. Our designs build upon the U-Net structure utilized in Denoising Diffusion Probabilistic Models (DDPM) Ho et al. (2020), incorporating advanced conditioning and attention mechanisms to enhance performance. For detailed implementation of the Noise Variance Conditioned Multi-Head Self-Attention (NVC-MSA) module, please refer to Appendix C.2.

### C.1   SINGLE-CONTRAST MODEL ARCHITECTURE

Our single-contrast denoising model employs a U-Net backbone augmented with time conditioning and the NVC-MSA module Hatamizadeh et al. (2023).

**Time Conditioning**: The model adapts its processing based on the noise level $t$ by integrating time embeddings into the convolutional layers. This is achieved through adaptive normalization (e.g., instance normalization followed by an affine transformation conditioned on the time embedding), as introduced in DDPM.

**NVC-MSA Module**: To enable the network to adjust to varying noise levels, we incorporate the NVC-MSA module into the self-attention mechanisms of the U-Net. The module conditions the attention on the current noise variance, allowing the network to effectively capture long-range dependencies and adapt to different noise scales. Mathematically, the queries, keys, and values are computed as:

$$\mathbf{Q} = \mathbf{W}_Q(\mathbf{X}) + \mathbf{b}_Q(t), \tag{28}$$
$$\mathbf{K} = \mathbf{W}_K(\mathbf{X}) + \mathbf{b}_K(t), \tag{29}$$
$$\mathbf{V} = \mathbf{W}_V(\mathbf{X}) + \mathbf{b}_V(t), \tag{30}$$

where $\mathbf{b}_Q(t)$, $\mathbf{b}_K(t)$, and $\mathbf{b}_V(t)$ are learned affine transformations of the time embedding. This NVC-MSA mechanism allows the attention modules to be aware of the noise level and adjust their focus accordingly, effectively capturing long-range dependencies. The implementation details and pseudo-code for the NVC-MSA module are provided in Appendix C.2.

### C.2   NOISE VARIANCE CONDITIONED MULTI-HEAD SELF-ATTENTION (NVC-MSA) PSEUDO CODE

Below is the code snippet of the NVC-MSA module, which is integral to both single-contrast and multi-contrast model architectures. This module conditions the self-attention mechanism on the noise variance level, enabling the model to adapt its attention based on the current noise level.

The code snippet encapsulates the core functionality of the NVC-MSA module. The module first normalizes the input tensor and generates queries, keys, and values for spatial tokens. It then reshapes and projects the noise variance embeddings using 1x1 convolutions. These noise-conditioned components are added to the queries, keys, and values before applying the attention mechanism. Finally, the output is rearranged and projected to produce the final feature map.

```python
def forward(self, x, noise_emb):

    # Get shape of input tensor
    b, c, h, w = x.shape
    n = h * w

    # Normalize the input tensor
    x = self.norm(x)

    # Generate queries, keys, and values for spatial tokens
    qkv = self.to_qkv(x).chunk(3, dim=1)
    q, k, v = map(lambda t: rearrange(t, 'b (h d) x y -> b (x y) h d', h=self.
        heads), qkv)

    # Reshape and project noise variance embeddings using 1x1 convolutions
    noise_emb = noise_emb.view(b, -1, 1, 1)
    noise_q = self.noise_query_conv(noise_emb)
    noise_k = self.noise_key_conv(noise_emb)
    noise_v = self.noise_value_conv(noise_emb)

    # Rearrange the projected noise variance embeddings
    noise_q = rearrange(noise_q, 'b (h d) x y -> b (x y) h d', h=self.heads)
    noise_k = rearrange(noise_k, 'b (h d) x y -> b (x y) h d', h=self.heads)
    noise_v = rearrange(noise_v, 'b (h d) x y -> b (x y) h d', h=self.heads)

    # Add noise variance-dependent components to queries, keys, and values
    q = q + noise_q
    k = k + noise_k
    v = v + noise_v

    # Apply attention mechanism
    out = self.attend(q, k, v)

    # Rearrange and project the output
    out = rearrange(out, 'b (x y) h d -> b (h d) x y', x=h, y=w)

    return self.to_out(out)
```

Listing 1: Code snippet for NVC-MSA implementation

## D   MULTI-CONTRAST C2S

### D.1   MULTI-CONTRAST MODEL ARCHITECTURE

The multi-contrast denoising model extends the single-contrast architecture to handle multiple input contrasts, thereby enhancing denoising performance by leveraging complementary information.

**Multicontrast Fusion**: The model accepts multiple contrast inputs by concatenating the primary and complementary contrast images (e.g., (T1, T2)). An initial convolution layer extracts feature embeddings from the fused contrasts, which are then processed through the U-Net architecture.

**NVC-MSA Module**: Similar to the single-contrast model, the multi-contrast model integrates the NVC-MSA module into its self-attention mechanisms. By conditioning the attention on the noise variance level $\sigma$, the

model can adaptively adjust its focus based on the current noise level, as detailed in the pseudo-code provided in Appendix C.2.

**Output Head**: Following the U-Net processing, the output head generates a single-channel image representing the denoised primary contrast image.

**Flexibility and Extensions**: The architecture can dynamically adjust to accommodate any number of input contrasts by modifying the input layer accordingly. Although our current implementation is based on U-Net, the NVC-MSA mechanism is compatible with other architectures, such as Vision Transformers Dosovitskiy et al. (2020); Liu et al. (2021), where it can replace standard multi-head self-attention modules to enhance model complexity and performance. This extension remains an avenue for future research.

### D.2 MULTI-CONTRAST C2S ALGORITHM

The multi-contrast C2S framework extends the single-contrast algorithm to leverage complementary information from auxiliary contrast images. Given a collection of noisy multi-contrast training images

$$\{(\mathbf{X}_{t_{\text{data}},i}, \mathbf{C}_i)\}_{i=1}^N,$$

where $\mathbf{X}_{t_{\text{data}},i} \in \mathbb{R}^d$ is the noisy target contrast image and $\mathbf{C}_i \in \mathbb{R}^{d \times c}$ represents $c$ auxiliary noisy contrast images, our goal is to estimate the clean target contrast image $\mathbf{X}_0$ using both $\mathbf{X}_{t_{\text{data}}}$ and $\mathbf{C}$. In the multi-contrast setting, we focus on the case where $t_{\text{target}} = 0$ (i.e., $\sigma_{t_{\text{target}}} = 0$), aiming to directly estimate the MMSE estimator $\mathbb{E}[\mathbf{X}_0 \mid \mathbf{X}_{t_{\text{data}}}, \mathbf{C}]$. While the single-contrast C2S incorporates a detail refinement extension with a non-zero target noise level, we leave the exploration of such extensions and more advanced contrast fusion architectures for multi-contrast C2S as future work. For the implementation of the denoising function $\mathbf{D}_\theta(\mathbf{X}_\tau, \tau \mid \mathbf{C})$, the conditioning on auxiliary contrasts $\mathbf{C}$ is achieved through a CNN encoder architecture that extracts features from each auxiliary contrast image. These extracted features are then concatenated with the features from the target contrast image in the feature space, allowing the model to effectively integrate complementary information from all available contrasts. The concatenated features are subsequently processed through the U-Net backbone with NVC-MSA modules, as in the single-contrast case. Following the reparameterization strategy introduced in Section 3.1, we define a reparameterized function $\mathbf{D}_\theta(\mathbf{X}_\tau, \tau \mid \mathbf{C})$ as:

$$\mathbf{D}_\theta(\mathbf{X}_\tau, \tau \mid \mathbf{C}) = \lambda_{\text{out}}(\tau, \sigma_{t_{\text{target}}}) \, \mathbf{h}_\theta(\mathbf{X}_t, t \mid \mathbf{C}) + \lambda_{\text{skip}}(\tau, \sigma_{t_{\text{target}}}) \, \mathbf{X}_t, \tag{31}$$

where $\mathbf{X}_t = \mathbf{X}_{t_{\text{data}}} + \sigma_\tau \mathbf{Z}$, with $\mathbf{Z} \sim \mathcal{N}(\mathbf{0}, \mathbf{I}_d)$, and the coefficients remain consistent with the single-contrast case:

$$\lambda_{\text{out}}(\tau, \sigma_{t_{\text{target}}}) = \frac{\sigma_\tau^2}{\sigma_\tau^2 + \sigma_{t_{\text{data}}}^2 - \sigma_{t_{\text{target}}}^2}, \quad \lambda_{\text{skip}}(\tau, \sigma_{t_{\text{target}}}) = \frac{\sigma_{t_{\text{data}}}^2 - \sigma_{t_{\text{target}}}^2}{\sigma_\tau^2 + \sigma_{t_{\text{data}}}^2 - \sigma_{t_{\text{target}}}^2} \tag{32}$$

The multi-contrast C2S loss function is then formulated as:

$$\mathcal{L}_{\text{MC-C2S}}(\theta) = \frac{1}{2} \mathbb{E}_{\substack{\mathbf{X}_{t_{\text{data}}} \sim p_{t_{\text{data}}}(\mathbf{x}) \\ \mathbf{C} \sim p(\mathbf{c}) \\ \tau \sim \mathcal{U}[0,T] \\ \mathbf{Z} \sim \mathcal{N}(\mathbf{0}, \mathbf{I}_d)}} \left[ w(\tau) \left\| \mathbf{D}_\theta(\mathbf{X}_\tau, \tau \mid \mathbf{C}) - \mathbf{X}_{t_{\text{data}}} \right\|_2^2 \right] \tag{33}$$

The complete training procedure for multi-contrast C2S is outlined in Algorithm 2.

---

**Algorithm 2** Multi-Contrast Corruption2Self Training Procedure

---

**Require:** Noisy multi-contrast dataset $\{(\mathbf{X}^i_{t_\text{data}}, \mathbf{C}^i)\}^N_{i=1}$, $\mathbf{h}_\theta$, max noise level $T$, batch size $m$, total iterations $K$, noise schedule function $\sigma_\tau$

1: **for** $k = 1$ to $K$ **do**
2:    Sample minibatch $\{(\mathbf{X}^i_{t_\text{data}}, \mathbf{C}^i)\}^m_{i=1}$, $\tau \sim \mathcal{U}(0, T]$, $\mathbf{Z} \sim \mathcal{N}(\mathbf{0}, \mathbf{I}_d)$
3:    Compute $\lambda_\text{out}(\tau, 0) = \frac{\sigma^2_\tau}{\sigma^2_\tau + \sigma^2_{t_\text{data}}}$, $\lambda_\text{skip}(\tau, 0) = \frac{\sigma^2_{t_\text{data}}}{\sigma^2_\tau + \sigma^2_{t_\text{data}}}$
4:    Recover $t = \sigma_t^{-1} \left( \sqrt{\sigma^2_\tau + \sigma^2_{t_\text{data}}} \right)$
5:    Compute $\mathbf{X}_t \leftarrow \mathbf{X}_{t_\text{data}} + \sigma_\tau \mathbf{Z}$
6:    Compute loss: $\mathcal{L} = \frac{1}{2m} \sum^m_{i=1} w(\tau) \left\| \lambda_\text{out}(\tau, 0) \, \mathbf{h}_\theta(\mathbf{X}^i_t, t \mid \mathbf{C}^i) + \lambda_\text{skip}(\tau, 0) \, \mathbf{X}^i_t - \mathbf{X}^i_{t_\text{data}} \right\|^2$
7:    Update $\theta$ using Adam optimizer to minimize $\mathcal{L}$

---

During inference, given a noisy target contrast observation $\mathbf{X}_{t_\text{data}}$ and auxiliary contrasts $\mathbf{C}$, the denoised output is obtained by:

$$\hat{\mathbf{X}} = \mathbf{h}_{\theta^*}(\mathbf{X}_{t_\text{data}}, t_\text{data} \mid \mathbf{C}) \tag{34}$$

where the trained model $\mathbf{h}_{\theta^*}$ approximates $\mathbb{E}[\mathbf{X}_0 \mid \mathbf{X}_{t_\text{data}}, \mathbf{C}]$, providing a clean estimate of the target contrast image that benefits from the complementary information in the auxiliary contrasts.

# E    ADDITIONAL RESULTS ON FASTMRI

In this section, we present additional results on the fastMRI dataset, evaluating the performance of various denoising methods across different noise levels and contrasts.

Table 6 summarizes the performance comparison between the baseline method (Without Reparameterization) and our proposed Corruption2Self (C2S) approach, under four configurations: PD with $\sigma = 13/255$, PDFS with $\sigma = 13/255$, PD with $\sigma = 25/255$, and PDFS with $\sigma = 25/255$.

| Method | **PD**, $\sigma = 13/255$ PSNR ↑ / SSIM ↑ | **PDFS**, $\sigma = 13/255$ PSNR ↑ / SSIM ↑ | **PD**, $\sigma = 25/255$ PSNR ↑ / SSIM ↑ | **PDFS**, $\sigma = 25/255$ PSNR ↑ / SSIM ↑ |
|---|---|---|---|---|
| Without Reparam. | 32.65 / 0.821 | 30.08 / 0.676 | 30.16 / 0.747 | 28.24 / 0.570 |
| With Reparam. | **33.48 / 0.832** | **30.67 / 0.761** | **30.91 / 0.756** | **28.62 / 0.601** |

Table 6: Impact of reparameterization of noise levels on the fastMRI dataset. The "Without Reparam." row contains estimated baseline results, while "With Reparam." represents the proposed method with reparameterization.

# F    ADDITIONAL RESULTS ON M4RAW

## F.1    VISUAL COMPARISON OF DENOISING METHODS

In this section, we provide a visual comparison of several denoising methods applied to T1, T2, and FLAIR contrast images in the M4Raw dataset. The denoising methods evaluated include Noise2Noise, BM3D, SwinIR, R2R, Noise2Self, and C2S, with Multi-repetition Averaged Label serving as the ground truth reference for comparison.

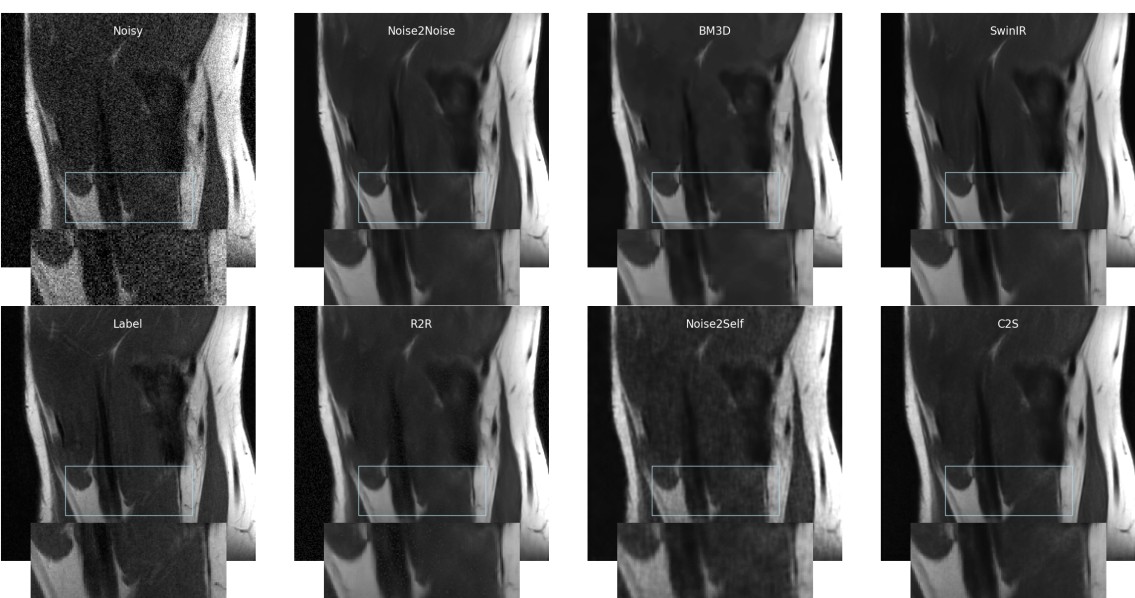

Figure 5: Comparison of different denoising methods for PD contrast (noise level 13/255) in fastMRI.

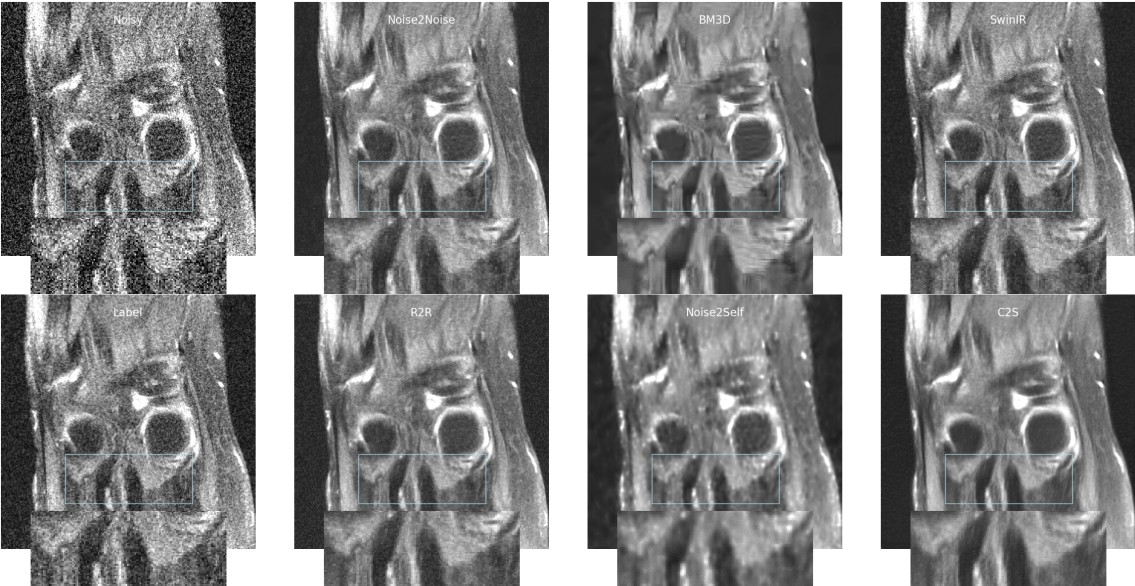

Figure 6: Comparison of different denoising methods for PDFS contrast (noise level 25/255) in fastMRI.

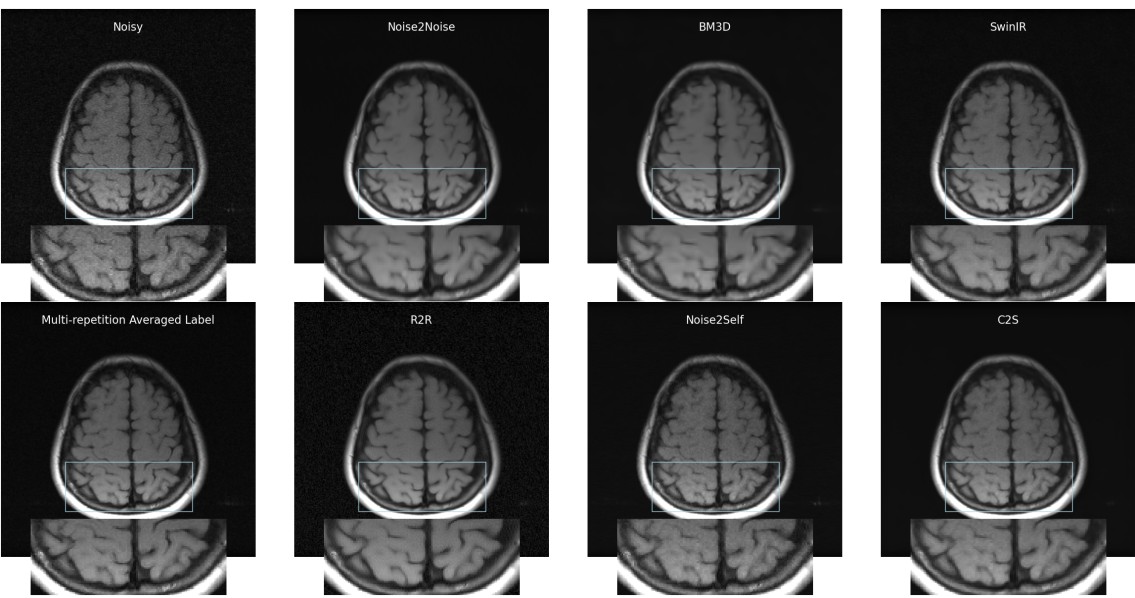

Figure 7: Comparison of different denoising methods for T1 contrast in M4Raw. The top row shows the original noisy image and results from Noise2Noise, BM3D, and SwinIR. The bottom row includes the multi-repetition averaged label, R2R, Noise2Self, and C2S methods. A zoomed-in section of each image is presented below each corresponding brain image for detailed comparison.

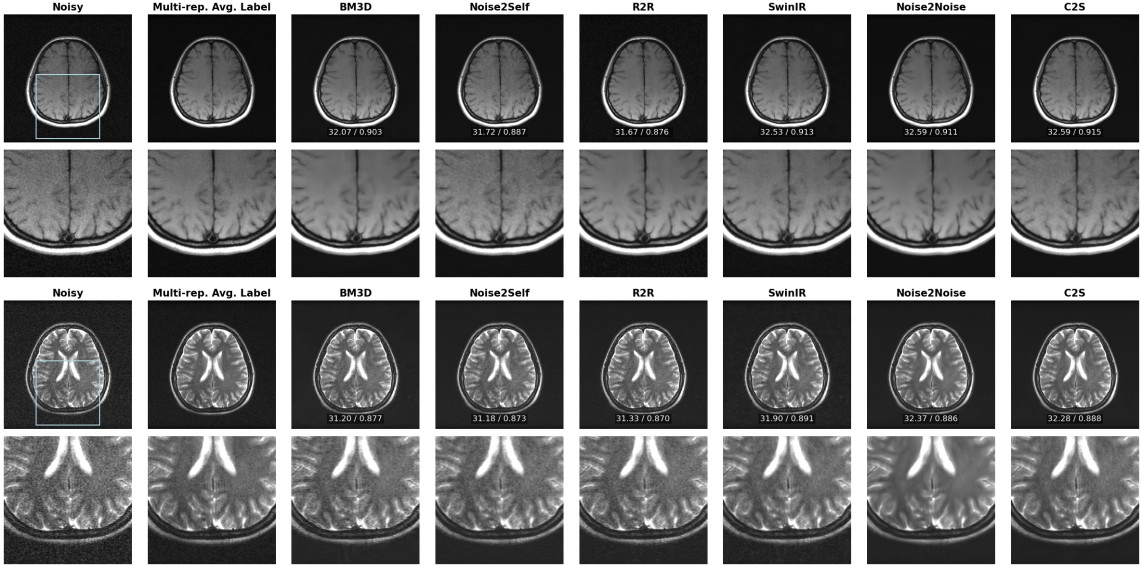

Figure 8: Top: Comparison of different denoising methods for T1 contrast in M4Raw. Bottom: Comparison of different denoising methods for T2 contrast in M4Raw.

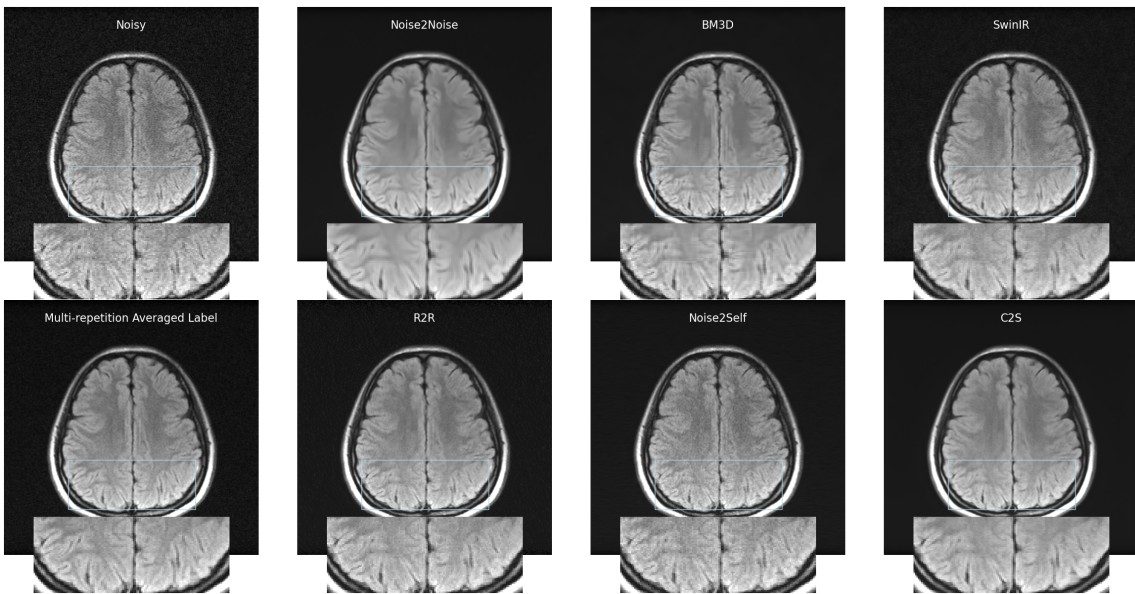

Figure 9: Comparison of different denoising methods for FLAIR contrast in M4Raw.

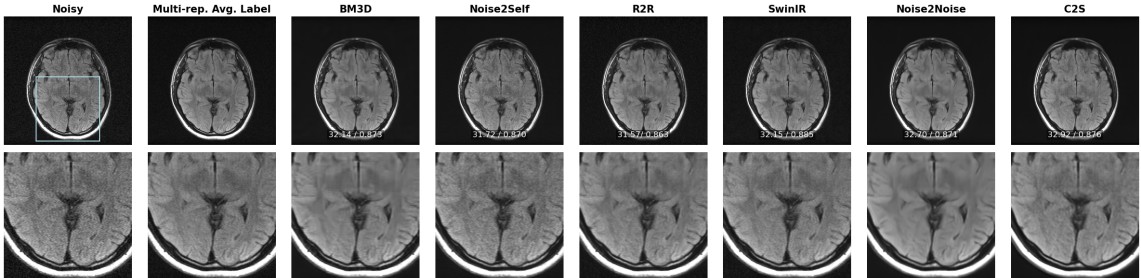

Figure 10: Comparison of different denoising methods for FLAIR contrast in M4Raw.

## F.2 EVALUATION ON MATCHING-SNR TEST LABELS

When evaluated on test data matching the SNR of the training data, supervised methods like SwinIR and Restormer achieve the best performance, with PSNR improvements over self-supervised methods. This indicates that supervised approaches excel when the training and testing conditions are similar. However, the performance gap narrows when considering C2S, which attains PSNR and SSIM values comparable to those of the supervised methods.

These observations highlight a limitation of supervised methods: they may not generalize well to scenarios where the test data has different noise characteristics or higher SNR than the training data. In contrast, C2S demonstrates robust performance across different SNR levels, achieving competitive results on both matching-SNR and higher-SNR test labels. This suggests that our self-supervised approach is more resilient to variations in noise levels and better generalizes to cleaner images without relying on clean ground truth during training.

| Methods | T1 PSNR / SSIM ↑ | T2 PSNR / SSIM ↑ | FLAIR PSNR / SSIM ↑ |
|---|---|---|---|
| *Classical Non-Learning-Based Methods* | | | |
| NLM Froment (2014) | 34.65 / 0.897 | 33.72 / 0.873 | 32.83 / 0.830 |
| BM3D Mäkinen et al. (2020) | 35.27 / 0.900 | 34.01 / 0.875 | 33.22 / 0.841 |
| *Supervised Learning Methods* | | | |
| SwinIR Liang et al. (2021) | 36.09 / 0.926 | 34.57 / 0.902 | 34.34 / 0.909 |
| Restormer Zamir et al. (2022) | 35.96 / 0.926 | 34.13 / 0.898 | 34.21 / 0.908 |
| Noise2Noise Lehtinen et al. (2018) | 34.82 / 0.892 | 33.92 / 0.861 | 33.73 / 0.879 |
| *Self-Supervised Single-Contrast Methods* | | | |
| Noise2Void Krull et al. (2019) | 32.83 / 0.870 | 31.73 / 0.857 | 30.90 / 0.821 |
| Noise2Self Batson & Royer (2019) | 34.17 / 0.883 | 32.64 / 0.847 | 31.96 / 0.823 |
| Recorrupted2Recorrupted Pang et al. (2021) | 33.60 / 0.801 | 32.93 / 0.820 | 32.42 / 0.794 |
| **C2S (Ours)** | **36.11 / 0.925** | **34.87 / 0.904** | **34.15 / 0.898** |

Table 7: Quantitative results on the M4Raw dataset evaluated on three-repetition-averaged test labels (matching training and validation SNR). Mean PSNR and SSIM metrics are reported. The best results among self-supervised methods are in bold.

## G  DETAIL REFINEMENT ALGORITHM

In this appendix, we present the **Detail Refinement** algorithm as an extension to the primary Corruption2Self (C2S) training procedure. While the primary stage focuses on learning the conditional expectation $\mathbb{E}[\mathbf{X}_0 \mid \mathbf{X}_t]$ for maximum denoising, this aggressive approach may lead to the loss of fine details and important features. The Detail Refinement stage addresses this issue by training the network to predict $\mathbb{E}[\mathbf{X}_{t_{\text{target}}} \mid \mathbf{X}_t]$ with a non-zero target noise level $\sigma_{t_{\text{target}}} > 0$, allowing the preservation of intricate structures and textures. Empirically, we discovered that uniformly sampling $t_{\text{target}}$ from the interval $[0, t_{\text{data}}]$ during training already yields good results and leave more advanced tuning process to future work. This sampling strategy provides a good balance between noise reduction and detail preservation, allowing the network to learn a range of refinement levels adaptively.

### G.1  LOSS FUNCTION FOR DETAIL REFINEMENT

The loss function for the Detail Refinement stage is similar to the primary C2S stage but introduces a target noise level $\sigma_{t_{\text{target}}}$. The denoised output $\mathbf{D}_\theta(\mathbf{X}_\tau, \tau, \sigma_{t_{\text{target}}})$ is defined as:

$$\mathbf{D}_\theta(\mathbf{X}_\tau, \tau, \sigma_{t_{\text{target}}}) = \lambda_{\text{out}}(\tau, \sigma_{t_{\text{target}}}) \, \mathbf{h}_\theta(\mathbf{X}_t, t) + \lambda_{\text{skip}}(\tau, \sigma_{t_{\text{target}}}) \, \mathbf{X}_t,$$

where $\mathbf{h}_\theta(\mathbf{X}_t, t)$ is the denoising network parameterized by $\theta$, with input $\mathbf{X}_t$ and noise level $t$, and $\lambda_{\text{out}}(\tau, \sigma_{t_{\text{target}}})$ and $\lambda_{\text{skip}}(\tau, \sigma_{t_{\text{target}}})$ are blending factors between the network output and the noisy input.

The loss function is given by:

$$\mathcal{L}_{\text{refine}}(\theta) = \frac{1}{2} \mathbb{E}_{\substack{\mathbf{X}_{t_{\text{data}}} \sim p_{t_{\text{data}}}(\mathbf{x}) \\ \tau \sim \mathcal{U}[0,T] \\ \sigma_{t_{\text{target}}} \sim \mathcal{U}(0, \sigma_{t_{\text{data}}}] \\ \mathbf{Z} \sim \mathcal{N}(\mathbf{0}, \mathbf{I}_d)}} \left[ w(\tau) \left\| \mathbf{D}_\theta(\mathbf{X}_\tau, \tau, \sigma_{t_{\text{target}}}) - \mathbf{X}_{t_{\text{data}}} \right\|_2^2 \right], \tag{35}$$

where:

- $\mathbf{X}_{t_{\text{data}}} \sim p_{t_{\text{data}}}(\mathbf{x})$ is the noisy observed data.
- $\tau \sim \mathcal{U}[0, T]$ is the reparameterized noise level uniformly sampled from $[0, T]$, where $T$ is the maximum noise level.
- $\sigma_{t_{\text{target}}} \sim \mathcal{U}(0, \sigma_{t_{\text{data}}}]$ is the target noise level, sampled uniformly from the interval $(0, \sigma_{t_{\text{data}}}]$.
- $\mathbf{X}_t = \mathbf{X}_{t_{\text{data}}} + \sigma_\tau \mathbf{Z}$, where $\mathbf{Z} \sim \mathcal{N}(\mathbf{0}, \mathbf{I}_d)$ is Gaussian noise.
- $\lambda_{\text{out}}(\tau, \sigma_{t_{\text{target}}})$ and $\lambda_{\text{skip}}(\tau, \sigma_{t_{\text{target}}})$ are defined as:

$$\lambda_{\text{out}}(\tau, \sigma_{t_{\text{target}}}) = \frac{\sigma_\tau^2}{\sigma_\tau^2 + \sigma_{t_{\text{data}}}^2 - \sigma_{t_{\text{target}}}^2}, \quad \lambda_{\text{skip}}(\tau, \sigma_{t_{\text{target}}}) = \frac{\sigma_{t_{\text{data}}}^2 - \sigma_{t_{\text{target}}}^2}{\sigma_\tau^2 + \sigma_{t_{\text{data}}}^2 - \sigma_{t_{\text{target}}}^2}. \tag{36}$$

The goal of this refinement stage is to retain a controlled amount of noise, preventing the loss of fine details and features while still performing denoising.

## G.2 ALGORITHM IMPLEMENTATION

The Detail Refinement training procedure minimizes the loss function $\mathcal{L}_{\text{refine}}(\theta)$ over the network parameters $\theta$. The steps are as follows:

---

**Algorithm 3** Detail Refinement Training Procedure

---

**Require:** Noisy dataset $\{\mathbf{X}_{t_{\text{data}}}^i\}_{i=1}^N$, denoising network $\mathbf{h}_\theta$, max noise level $T$, batch size $m$, total iterations $K_{\text{refine}}$, noise schedule function $\sigma_\tau$

1: **for** $k = 1$ to $K_{\text{refine}}$ **do**
2:      Sample minibatch $\{\mathbf{X}_{t_{\text{data}}}^i\}_{i=1}^m$, $\tau \sim \mathcal{U}(0, T]$, $\mathbf{Z} \sim \mathcal{N}(\mathbf{0}, \mathbf{I}_d)$
3:      Sample $\sigma_{t_{\text{target}}} \sim \mathcal{U}(0, \sigma_{t_{\text{data}}}]$
4:      Compute $\lambda_{\text{out}}(\tau, \sigma_{t_{\text{target}}}) = \frac{\sigma_\tau^2}{\sigma_\tau^2 + \sigma_{t_{\text{data}}}^2 - \sigma_{t_{\text{target}}}^2}$
5:      Compute $\lambda_{\text{skip}}(\tau, \sigma_{t_{\text{target}}}) = \frac{\sigma_{t_{\text{data}}}^2 - \sigma_{t_{\text{target}}}^2}{\sigma_\tau^2 + \sigma_{t_{\text{data}}}^2 - \sigma_{t_{\text{target}}}^2}$
6:      Recover $t = \sigma_t^{-1}\left(\sqrt{\sigma_\tau^2 + \sigma_{t_{\text{data}}}^2}\right)$
7:      Compute $\mathbf{X}_t \leftarrow \mathbf{X}_{t_{\text{data}}} + \sigma_\tau \mathbf{Z}$
8:      Compute loss: $\mathcal{L} = \frac{1}{2m} \sum_{i=1}^m w(\tau) \left\| \mathbf{D}_\theta(\mathbf{X}_\tau, \tau, \sigma_{t_{\text{target}}}) - \mathbf{X}_{t_{\text{data}}}^i \right\|^2$
9:      Update $\theta$ using Adam optimizer Kingma (2014) to minimize $\mathcal{L}$

---

# H ROBUSTNESS TO NOISE LEVEL ESTIMATION ERROR

In practical applications, precise noise level estimation can be challenging, making robustness to estimation errors a crucial property for denoising models. This section provides a comprehensive analysis of our model's performance under various degrees of noise level misestimation, demonstrating its capability to function effectively as a blind denoising model.

## H.1 EXPERIMENTAL SETUP AND RESULTS

We evaluate our pretrained model's robustness by systematically varying the input noise level estimation $t_{\text{data}}$ from -50% (underestimation) to +50% (overestimation) of the true noise level. Table 8 presents the quantitative results across T1, T2, and FLAIR contrasts on the M4Raw test set, while Figure 11 visualizes these trends.

| Noise Level | M4Raw Dataset (PSNR / SSIM ↑) | | |
| Estimation | T1 | T2 | FLAIR |
| --- | --- | --- | --- |
| -50% | 32.6004 / 0.9154 | 32.3210 / 0.8890 | 32.9496 / 0.8757 |
| -40% | 32.5958 / 0.9153 | 32.3125 / 0.8888 | 32.9498 / 0.8759 |
| -30% | 32.5910 / 0.9151 | 32.3044 / 0.8886 | 32.9478 / 0.8761 |
| -20% | 32.5861 / 0.9150 | 32.2964 / 0.8884 | 32.9436 / 0.8762 |
| -10% | 32.5810 / 0.9149 | 32.2882 / 0.8881 | 32.9373 / 0.8763 |
| 0% | 32.5758 / 0.9148 | 32.2812 / 0.8879 | 32.9297 / 0.8764 |
| +10% | 32.5704 / 0.9147 | 32.2700 / 0.8876 | 32.9192 / 0.8763 |
| +20% | 32.5649 / 0.9146 | 32.2593 / 0.8873 | 32.9076 / 0.8763 |
| +30% | 32.5592 / 0.9145 | 32.2468 / 0.8870 | 32.8947 / 0.8762 |
| +40% | 32.5535 / 0.9143 | 32.2322 / 0.8866 | 32.8805 / 0.8760 |
| +50% | 32.5475 / 0.9142 | 32.2151 / 0.8862 | 32.8655 / 0.8759 |

Table 8: Performance analysis under varying noise level estimations on the M4Raw dataset. The model demonstrates remarkable stability across all contrasts, with particularly strong performance under slight underestimation. The '-' and '+' indicate underestimation and overestimation of the noise level, respectively.

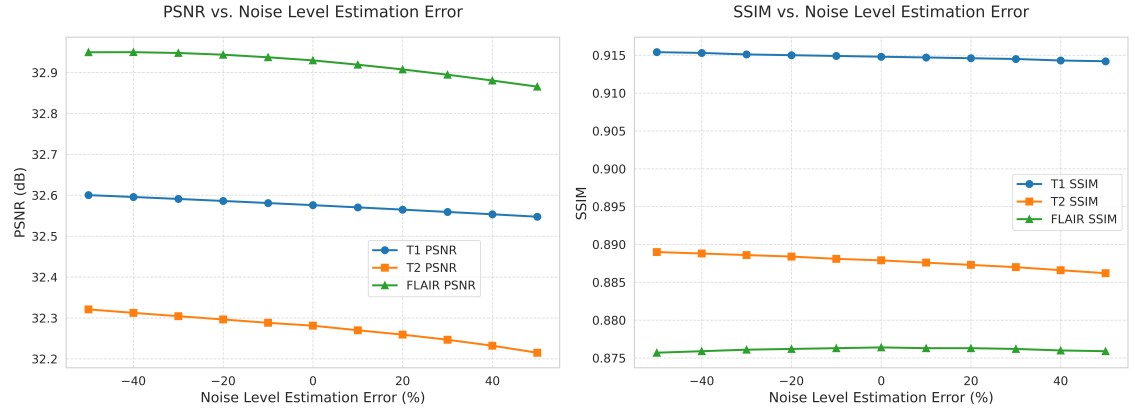

Figure 11: Visualization of model performance under varying noise level estimations. The plots demonstrate consistent stability across all contrasts, with minimal performance degradation even under significant estimation errors (±50%).

## H.2 ANALYSIS AND DISCUSSION

Our experiments reveal several key findings: **Overall Stability**: The model maintains remarkably stable performance across all tested estimation errors, with maximum PSNR variations of only 0.053 dB, 0.106 dB, and 0.084 dB for T1, T2, and FLAIR contrasts, respectively. **Asymmetric Response**: Interestingly, the model shows slightly better performance under noise level underestimation compared to overestimation. For instance, with T1 contrast, a 50% underestimation achieves a PSNR of 32.6004 dB, outperforming both the true noise level (32.5758 dB) and 50% overestimation (32.5475 dB). While all contrasts exhibit robust performance, the degree of stability varies. T1 shows the most stable response, while T2 demonstrates slightly higher sensitivity to estimation errors.

| $T$ | M4Raw T1 | | fastMRI PD25 | |
|---|---|---|---|---|
| | PSNR ↑ | SSIM ↑ | PSNR ↑ | SSIM ↑ |
| 3 | 33.30 | 0.869 | 30.58 | 0.740 |
| 5 | 33.91 | 0.879 | **31.01** | **0.765** |
| 10 | **34.91** | **0.890** | 30.41 | 0.751 |
| 15 | 34.35 | 0.882 | 30.42 | 0.743 |
| 20 | 34.43 | 0.882 | 30.52 | 0.755 |

Table 9: Influence of maximum corruption level $T$ on the M4Raw validation dataset.

## H.3 PRACTICAL IMPLEMENTATION

In our implementation, we utilize the `skimage` package Van der Walt et al. (2014) for noise level estimation, which proves sufficient for optimal performance. The model's demonstrated robustness suggests that even relatively simple estimation techniques can provide adequate noise level approximations for effective denoising. Our findings reveal significant practical advantages: the model readily adapts to real-world scenarios where exact noise levels are unknown, while standard noise estimation tools consistently deliver near-optimal performance. Furthermore, the observed slight preference for underestimation indicates that conservative noise level estimates may be advantageous in practice.

While future work could explore more sophisticated noise estimation techniques, particularly for extreme cases, our current results demonstrate that the model's inherent robustness already makes it highly practical for real-world applications. This robustness, combined with the effectiveness of standard noise estimation tools, enables our approach to function reliably as a blind denoising model, requiring minimal assumptions about the underlying noise characteristics.

## I ADDITIONAL ABLATION STUDIES

### I.1 EFFECTIVENESS OF MAXIMUM CORRUPTION LEVEL $T$

We also analyzed the impact of the maximum corruption level $T$. All models were trained for 300 epochs with the same hyperparameters. As shown in Table 9, performance generally improves with higher $T$, peaking at $T = 10$ for the M4Raw dataset (T1 contrast). For the fastMRI dataset (PD contrast, 25/255 noise level), the best performance occurs at $T = 5$. These results indicate that while increasing $T$ generally benefits performance, excessively high corruption levels may not lead to further improvements within the given training budget and could require longer training times to converge.

### I.2 EFFECTIVENESS OF REPARAMETERIZATION AND EMA ON TRAINING DYNAMICS

In our framework, we leverage a reparameterization strategy to address the challenges associated with noise level sampling. The theoretical foundation of GDSM (see Theorem 1) requires sampling noise levels $t > t_{\text{data}}$. However, directly sampling $t \sim \mathcal{U}(t_{\text{data}}, T]$ can be sensitive to errors in noise level estimation. To mitigate this, we instead sample an auxiliary variable $\tau \sim \mathcal{U}(0, T]$ and define the corresponding variance via

$$\sigma_\tau^2 = \sigma_t^2 - \sigma_{t_{\text{data}}}^2.$$

We now describe two key perspectives that elucidate why this reparameterization improves the training process:

**Stable Noise Level Sampling.** By recovering the original noise level through the transformation

$$t = \sigma_t^{-1}\left(\sqrt{\sigma_\tau^2 + \sigma_{t_{\text{data}}}^2}\right),$$

the additive term $\sigma_\tau^2$ ensures that the recovered $t$ remains valid even when $\sigma_{t_{\text{data}}}$ is underestimated. This robustness contrasts with the direct sampling approach, where inaccuracies in estimating $t_{\text{data}}$ would directly affect the noise level range.

**Consistent Coverage Across Data Samples.** Sampling $\tau$ from a fixed interval $(0, T]$ guarantees that the entire noise level range is consistently covered during training, irrespective of individual sample noise characteristics. Although we approximate $T \approx T'$ for practical implementation, the fixed-range sampling of $\tau$ avoids the variability inherent in the direct sampling of $t$ (which would otherwise depend on each sample's noise level). This consistency is particularly beneficial for datasets with heterogeneous noise levels, leading to smoother convergence and more stable training dynamics.

In addition to reparameterization, we apply an Exponential Moving Average (EMA) with a decay rate of 0.999 during training. The combination of these techniques not only enhances stability but also accelerates convergence. Figure 12 presents a comparative analysis of the PSNR and SSIM metrics on the M4Raw FLAIR dataset over the first 125 training epochs ($T = 5$ and $\sigma_{\text{target}} = 0$). The results demonstrate that the joint application of reparameterization and EMA stabilizes the training dynamics.

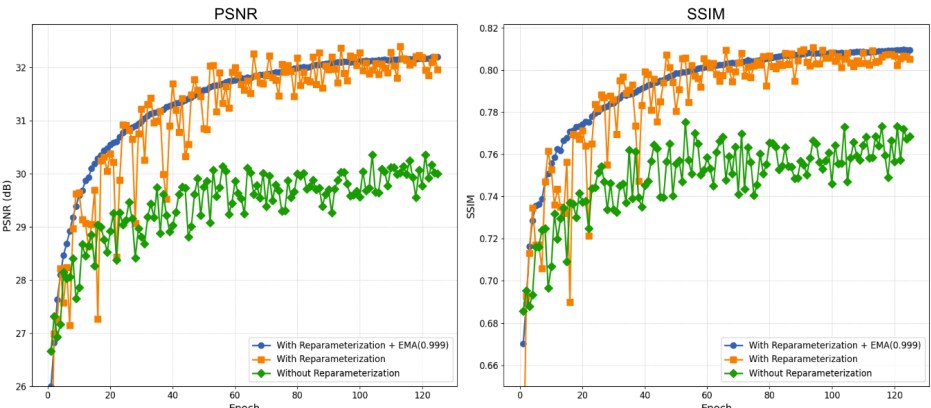

Figure 12: *Effectiveness of Reparameterization in Noise Level Adjustment on the M4Raw FLAIR validation Dataset.* Comparison of PSNR and SSIM metrics on the validation set for different model configurations over the first 125 training epochs ($T = 5$, $\sigma_{\text{target}} = 0$). The combination of reparameterization and EMA (0.999) yields more stable training dynamics and improved convergence.

## J GUIDELINES FOR SELECTING AND USING MAXIMUM CORRUPTION LEVEL

### J.1 THEORETICAL BOUNDS FOR MAXIMUM CORRUPTION LEVEL

Following insights from score-based generative models Song et al. (2020), the maximum corruption level $T$ can be theoretically chosen as large as the maximum Euclidean distance between all pairs of training data points to ensure sufficient coverage for accurate score estimation. Our analysis of MRI datasets reveals the following maximum pairwise distances:

- **M4Raw Dataset:**
  - T1 contrast: 56.72
  - T2 contrast: 47.99
  - FLAIR contrast: 65.20
- **FastMRI Dataset:**
  - PD ($\sigma = 13/255$): 114.23
  - PDFS ($\sigma = 13/255$): 102.59
  - PD ($\sigma = 25/255$): 115.53
  - PDFS ($\sigma = 25/255$): 102.94

## J.2 PRACTICAL CONSIDERATIONS AND TRAINING DYNAMICS

While theoretical bounds provide upper limits, our empirical studies show that significantly smaller values of $T$ can achieve optimal performance while maintaining computational efficiency. Table 10 demonstrates the relationship between $T$ and training convergence:

| Max Corruption Level ($T$) | Epochs to Converge |
| :---: | :---: |
| 20 | 204 |
| 15 | 183 |
| 10 | 125 |
| 5 | 79 |
| 3 | 42 |

Table 10: Relationship between maximum corruption level and training convergence on M4Raw dataset (T1).

## J.3 ALTERNATIVE: VARIANCE PRESERVING (VP) FORMULATION

An alternative to the variance exploding (VE) formulation is the variance preserving (VP) formulation (Appendix B.1). Here, $\sigma_t$ is sampled uniformly from $(\sigma_{t_{\text{data}}}, 1)$, eliminating the need to estimate $T$. This approach avoids explicit selection of $T$, as the maximum corruption level is bounded by 1.

The corruption process in this formulation is given by:

$$\mathbf{X}_{t_{\text{data}}} = \sqrt{1 - \sigma_{t_{\text{data}}}^2} \mathbf{X}_0 + \sigma_{t_{\text{data}}} \mathbf{Z}, \quad 0 < \sigma_{t_{\text{data}}} < 1, \tag{37}$$

with additional noise levels sampled such that $\sigma_{t_{\text{data}}} < \sigma_t < 1$. This principled approach maintains theoretical guarantees and achieves comparable performance to VE in our experiments.

## J.4 PRACTICAL GUIDELINES

Based on our analysis, we recommend starting with $T = 5$ for datasets exhibiting moderate noise levels, as this provides an effective baseline for most applications. For datasets with higher noise levels or more complex noise patterns, gradually increasing $T$ up to 20 may yield improved results, though practitioners should note that training time increases approximately linearly with $T$. Throughout the training process, careful monitoring of validation metrics is essential to determine optimal stopping points and assess the effectiveness of the chosen corruption level. In cases where dataset characteristics make the selection of $T$ particularly challenging, the VP formulation offers a principled alternative that maintains theoretical guarantees while eliminating the need for explicit $T$ selection.

