# OpenReview forum: "Score-based Self-supervised MRI Denoising"
_ICLR.cc/2025/Conference — ICLR 2025 Poster_

### Official Review · Reviewer_jWG5 · 2024-11-04

**Soundness:** 3
**Presentation:** 3
**Contribution:** 3
**Rating:** 8
**Confidence:** 4

**Summary:**

This paper proposes Corruption2Self, a score-based self-supervised framework for MRI denoising.

Motivation:

The goal is to learn denoising directly from noisy data (without relying on high-quality labels). Their framework aims to fix the label scarcity and over-smoothing of finer details issues in existing supervised and self-supervised methods respectively.

Contributions:

The framework comprises a generalized ambient denoising score matching (GADSM) loss followed by reparametrization to improve convergence and detail refinement extension to preserve finer spatial features. Further, the authors extend the framework to incorporate additional MRI contrasts to improve performance. The authors finally claim that their framework achieves state-of-the-art performance among self-supervised methods and comparable performance among supervised methods.

**Strengths:**

1. The paper comprehensively analyzes and applies existing self-supervised and supervised denoising approaches in the context of MRI

2. The paper is well written, original and provides great detail into the workflow of the Corruption2Self (C2F) framework.

3. Incorporation to reparametrization (Table 4) and extensions to multi-contrast settings to improve denoising.

4. Showcases robustness of methodology on varied noise level estimations compared to true noise (Table 9)

**Weaknesses:**

### A. Detail refinement extension claim
According to the metrics in Table 1, it is unclear if the detail refinement extension is being effective. The improvements in PSNR / SSIM does not seem notable. It would be helpful to include an error bar (for the table), statistical significance test (to show notability) and visuals to show effectiveness.

### B. Applicability and impact
The paper can cover how their workflow can be used in practice while denoising. The following aspects can add more value-add in terms of real-world impact to the paper.
1. Estimation of $σ_{tdata}$ in real-time during inference.
2. Given that traditional methods such as BM3D work decently well (maybe 1-3 points less performant than C2S), how difficult is it to go about training / deploying C2S in the MRI denoising workflow than using non-learning based methods?
3. Would this methodology potentially result in MRI image acquisition?

**Questions:**

1. Robustness of C2S (M4Raw):
In the context of matching test-train SNR on M4Raw dataset, the authors claim that
    1. (Line 327-328) supervised methods such as SwinIR and Restormer perform better when the noise characteristics of train and test data are similar
    2. (Line 328-330) C2S achieves better generalization.
According to
    1. Table 3 (Results where test data SNR > train data SNR): C2S perform similar to SwinIR for eg.
    2. Table 7 (Results where test data SNR ~ train data SNR): C2S perform similar to SwinIR (although both have higher metrics here).

    It seems that both methods perform similar to each other in both the conditions?
    Are the metrics higher in latter because the labels in Table 7 are noisier than in Table 3? I'm not sure if we can conclude that C2S is more generalized and supervised is not from the above data?

2. I'd be interested to know the intuition behind why reparametrization works. Mathematically, it seems very similar except that we sample ${\tau}$ ~ $U(0, T]$ rather than ${\tau}$ ~ $U(0, T']$ due to $T >> T'$ . Does this approximation help in convergence?

3. Does reparametrization and detail refinement extension help in the case of FastMRI dataset also? Curious as effectiveness for only M4Raw dataset have been reported.

4. How would one estimate `T` (max corruption level) before training the model?

5. Given that complementary pair of contrasts improve denoising, curious to know if all the 3 contrasts can be used (eg. T1, T2 and FLAIR) for instance to further improve denoising?

---

> ### Author Response · Authors · 2024-11-28
>
> Thank you for your thoughtful and constructive feedback.
>
> > W-A (Detail Refinement Extension Effectiveness):
>
> We appreciate the reviewer's concern about the detail refinement extension's effectiveness. We have expanded our analysis with additional quantitative and qualitative evidence:
>
> 1) We have updated Table 1 to include error bars and statistical significance tests for both PSNR and SSIM metrics on the M4Raw dataset. The improvements are statistically significant ($p < 0.05$) across all contrasts.
>
> 2) These improvements are further tested on the fastMRI dataset, as shown in Table 2, where the detail refinement extension consistently enhances performance across different noise levels and contrasts. For instance, with PD contrast at $\sigma$ = 13/255, we observe improvements in both PSNR (from 33.36 to 33.48) and SSIM (from 0.831 to 0.832). The visual impact of detail refinement is shown in our updated Figures 3 and 4. We have also expanded Appendix G with visual comparisons that highlight the extension's effectiveness in preserving fine details while maintaining noise suppression.
>
>
> > W-B (Practical Applicability and Clinical Impact):
>
> We address three key aspects of C2S's practical implementation:
>
> 1. Real-time Noise Level Estimation: Our framework incorporates scikit-image's noise variance estimation package as a lightweight, practical solution for real-time $\sigma_{t_{\text{data}}}$ estimation. Importantly, our studies in Appendix H demonstrate that C2S maintains stable performance even with estimation errors of ±50\%, with PSNR variations of only 0.053dB, 0.106dB, and 0.084dB for T1, T2, and FLAIR contrasts respectively. This robustness makes C2S reliable in clinical settings where precise noise estimation may be challenging.
>
> 2. Implementation Compared to Traditional Methods: While non-learning methods like BM3D require no training and are straightforward to deploy, they achieve 1-3dB lower PSNR than C2S (Table 2 and 3). Our framework requires one-time offline training but offers marked advantages in practice. The trained model can be efficiently deployed using standard deep learning frameworks, providing superior denoising performance and feature preservation. Furthermore, C2S can handle multiple contrasts and adapt to varying noise conditions, making it more versatile than traditional approaches while remaining computationally feasible for clinical deployment.
>
> 3. Impact on MRI Acquisition: As a post-processing method, C2S enables potential improvements in MRI acquisition protocols. The proposed SNR enhancement approach can help reduce scan times and enable higher-resolution imaging without compromising SNR, providing more flexible trade-offs between acquisition speed and image quality.
>
> >  Q1 (Performance Comparisons Across Different SNR Conditions):
>
> We appreciate the reviewer's careful analysis of our results. We acknowledge that our original claim about "better generalization" should be refined. Rather than claiming superior generalization, we emphasize that C2S achieves competitive performance to supervised methods across varying SNR conditions while eliminating the need for clean training data on the M4Raw dataset.
>
> > Q2 (Reparameterization Intuition):
>
> The key insight behind our reparameterization lies in how it transforms the noise level sampling process. Theoretically, GADSM requires sampling noise levels $t > t_{\text{data}}$ (Theorem 1). Rather than directly sampling $t \sim \mathcal{U}(t_{\text{data}}, T]$, which can be sensitive to noise estimation errors, we reparameterize through $\tau \sim \mathcal{U}(0, T]$ where $\sigma_\tau^2 = \sigma_t^2 - \sigma_{t_{\text{data}}}^2$. We provide two perspectives to explain why our reparameterization strategy helps improve the training process:
>
> First, it ensures more stable noise level sampling. When we recover the original time via $t = \sigma_t^{-1}(\sqrt{\sigma_\tau^2 + \sigma_{t_{\text{data}}}^2})$, the addition of $\sigma_\tau^2$ helps maintain valid noise levels even when $\sigma_{t_{\text{data}}}$ might be underestimated. This approach is more robust compared to direct sampling from an estimated $(t_{\text{data}}, T]$ interval.
>
> Second, it enables consistent coverage of the noise level range during training. While the approximation $T \approx T'$ simplifies implementation, the key benefit comes from sampling $\tau$ from a fixed range $(0, T]$. This provides consistency across different data samples regardless of their individual noise levels - particularly important when the dataset contains varying noise levels, as the original sampling range would fluctuate with each sample's noise level. As demonstrated in Figure 2, this leads to smoother convergence and more stable training dynamics.

---

> > ### Author Response · Authors · 2024-11-28
> >
> > > Q3 (Effectiveness on fastMRI Dataset):
> >
> > We have included results for both reparameterization and detail refinement on the fastMRI dataset, complementing our results on M4Raw. Table 9 (Appendix E) shows consistent improvements across all contrasts and noise levels with our reparameterization strategy. We also updated Table 3 to show the effectiveness of the detail refinement extension on the fastMRI dataset.
> >
> > > Q4 (Selection of Maximum Corruption Level):
> >
> > We acknowledge that determining the exact optimal maximum corruption level T a priori remains an open challenge. While we cannot currently provide a method for estimating the optimal T before training, we can offer both theoretical insights and practical guidelines below.
> >
> > From a theoretical perspective, following insights from score-based generative models[1], T should ideally be as large as the maximum Euclidean distance between training data pairs to ensure sufficient coverage for accurate score estimation. In our analysis of MRI datasets, we observed maximum pairwise distances ranging from 47.99 to 65.20 for M4Raw contrasts and 102.59 to 115.53 for fastMRI contrasts. However, our empirical studies reveal that significantly smaller values of T can achieve optimal performance while maintaining computational efficiency.
> >
> > To address this challenge, we propose two practical solutions. In our variance exploding (VE) formulation, our empirical studies in Section H demonstrate that starting with T = 5 for datasets with moderate noise levels provides a good baseline, with the option to increase up to T = 10 for higher noise scenarios. This range offers an effective balance between performance and training efficiency. For practitioners who prefer to avoid T selection entirely, we also present a variance preserving (VP) formulation (detailed in Section B.1) where $\sigma_t$ is sampled uniformly from $(\sigma_{t_{\text{data}}}, 1)$. This alternative approach alleviates the need to estimate T while achieving comparable denoising performance.
> >
> > > Q5 (Three-Contrast Denoising Performance):
> >
> > Utilizing all three contrasts (T1, T2, and FLAIR) simultaneously indeed yields further improvements in denoising performance. As shown in Table 7 (Appendix D), our tri-contrast C2S achieves the best results across all metrics, outperforming both single-contrast and dual-contrast variants. For T1 contrast, the PSNR improves from 33.57dB (T1, T2) and 34.11dB (FLAIR, T1) to 34.18dB when using all three contrasts. Similar improvements are observed for T2 (33.63dB vs. 33.44dB) and FLAIR (33.07dB vs. 33.02dB) contrasts. We also included implementation specifics and additional results in Appendix D.2.
> >
> >
> > [1].Yang Song and Stefano Ermon. Improved techniques for training score-based generative models. Advances
> > in neural information processing systems, 33:12438–12448, 2020.

---

> > > ### Comment · Reviewer_jWG5 · 2024-12-02
> > >
> > > Thank you addressing my questions. I am increasing the soundness to 3 and sticking with the overall score of 8.
> > > I think this paper is a good application of score-based models for MRI denoising and I believe that the community could benefit from it.

---

> > > > ### Author Response · Authors · 2024-12-03
> > > >
> > > > Thank you for engaging with our responses and for your thorough evaluation throughout the review process! Your constructive feedback has helped strengthen this work.

---

### Official Review · Reviewer_9qMe · 2024-11-04

**Soundness:** 3
**Presentation:** 3
**Contribution:** 2
**Rating:** 6
**Confidence:** 4

**Summary:**

This paper proposed Corruption2Self, a self-supervised MRI denoising method based on ambient diffusion (training diffusion models using corrupted data). The authors proposed an algorithm named Reparametrized Generalized Ambient Denoising Score Matching and showed its superior performance compared to a number of baselines, supervised and self-supervised methods, on M4Raw (containing pairs of real noise and ground truth) and FastMRI datasets (synthetic noisy images).

**Strengths:**

- The proposed algorithm GADSM has sound math groundings.
- Authors did extensive experiments to compare the proposed algorithm to a number of baselines and showed its superior performance.
- Authors discussed the application of the algorithm to multi-contrast MRI, which is an overlooked field in MRI denoising.

**Weaknesses:**

- This paper's originality seems to be limited. The proposed method Reparametrized GADSM is a straightforward extension to ADSM [1]. In addition, authors failed to point out the challenges when applying self-supervised denoising methods in natural images to MRI images. It seems to be that except for the multi-contrast part, the others are natural extensions of techniques that have already been tested on natural images.
- The paper's problem setup is very similar to Noiser2noise [2], both handling Gaussian noise with known sigma. I think it is an important baseline to be tested.
- The self-supervised method used as baselines in this paper are too outdated. There are a number of newer methods under the category of blind-spot network (J-invariance) such as LG-BPN [3] and PUCA [4], which has more powerful architectures to increase the accuracy in predicting the value in blind spots, therefore better PSNR/SSIM numbers. Even though these methods were proposed to handle noise with spatial correlation (i.e. not pixelwise independent), the architectures can be easily optimized for independent noise (by making the blind spot be just 1 pixel).
- In Fig. 4, C2S seems to have more blurry results than R2R. Some details seem to be harder to see.
- The dataset shown in Fig. 4 seems to be unfair for supervised method, since the label is very noisy. Authors may want to re-consider the statement that "the potential of self-supervised learning to match or even surpass supervised methods in MRI denoising" (Introduction).
- The multi-contrast experiments are not fair to other baselines such as Noise2noise and R2R, since the other contrasts can be easily included as an extra channel in the model input to boost performance.
- Is hallucination problem of generative diffusion models a concern here? How can it be addressed?

[1] Daras G, Dimakis AG, Daskalakis C. Consistent Diffusion Meets Tweedie: Training Exact Ambient Diffusion Models with Noisy Data. arXiv preprint arXiv:2404.10177. 2024 Mar 20.
[2] Moran N, Schmidt D, Zhong Y, Coady P. Noisier2noise: Learning to denoise from unpaired noisy data. InProceedings of the IEEE/CVF Conference on Computer Vision and Pattern Recognition 2020 (pp. 12064-12072).
[3] Wang Z, Fu Y, Liu J, Zhang Y. Lg-bpn: Local and global blind-patch network for self-supervised real-world denoising. InProceedings of the IEEE/CVF Conference on Computer Vision and Pattern Recognition 2023 (pp. 18156-18165).
[4] Jang H, Park J, Jung D, Lew J, Bae H, Yoon S. PUCA: patch-unshuffle and channel attention for enhanced self-supervised image denoising. Advances in Neural Information Processing Systems. 2024 Feb 13;36.

**Questions:**

- Authors may want to point out clearly how MRI denoising is different from natural image denoising. What are the challenges? Why does it worths special attention?
- In the multi-contrast experiment, how extra contrasts were used as inputs? They were directly modeled by diffusion models (i.e. a channel of X_0 ... X_T) or as a conditional channel (i.e. diffusion models learn p(target contrast|other contrasts))?

---

> ### Author Response · Authors · 2024-11-28
>
> Thanks for your thoughtful and constructive feedback.
> > W1 (Limited Originality and MRI-Specific Challenges):
>
> While our work builds upon ADSM, we introduce several innovations specifically for self-supervised MRI denoising. To the best of our knowledge, we are the first to apply ADSM principles to create a self-supervised denoising framework that estimates the MMSE without clean labels, which is crucial in MRI applications where high-quality data are scarce or impractical to obtain. Our proposed score-based self-supervised denoising framework allows us to take advantage of the recent advances in diffusion model architectures. Additionally, our method incorporates a Noise Variance Conditioned Multi-Head Self-Attention (NVC-MSA) module (Appendix C) that dynamically adapts attention mechanisms based on noise levels, improving performance over classical DDPMs.
> Furthermore, we propse a detail refinement extension (Section 3.2, Appendix G) that addresses the need to preserve fine anatomical structures while reducing noise -- a crucial requirement for accurate clinical diagnosis where feature preservation is paramount for clinical applications and visual quality carries more weight than conventional metrics. We also introduce a noise level reparameterization scheme (Section 3.1) that ensures uniform sampling across noise levels, addressing the uneven sampling issue in the original ADSM framework and leading to demonstrably better convergence (Figure 2). Lastly, we provide extensive experimental validation on real and simulated MRI datasets (M4Raw and fastMRI), demonstrating practical evidence of our method’s effectiveness in real clinical settings.
>
> > W2 (Noisier2Noise):
>
> We appreciate the reviewer highlighting this connection. In fact, we demonstrate in Appendix B.2 that Noisier2Noise emerges as a special case of our GADSM framework under specific conditions. Specifically, when setting $\sigma_{t_{\text{target}}} = 0$ and fixing the noise ratio $\alpha^2 = (\sigma_t^2 - \sigma_{t_{\text{data}}}^2)/\sigma_{t_{\text{data}}}^2$, our framework reduces to Noisier2Noise's formulation (Equations 17-20). However, our approach offers several key advantages: (1) It provides a continuous noise schedule through $\sigma_t$, allowing the model to learn from a spectrum of noise levels rather than a fixed ratio determined by $\alpha$, (2) It introduces explicit time conditioning in the network architecture, enabling better adaptation to different noise magnitudes, and (3) Most importantly, it eliminates the need to tune the $\alpha$ parameter, which according to [1] is "difficult or impossible to derive in the absence of clean validation data" -- a typical situation in MRI applications where clean validation data is very limited or even unavailable. Furthermore, GADSM extends beyond clean image prediction by supporting arbitrary target noise levels through $\sigma_{t_{\text{target}}}$, providing a unified framework for various denoising objectives.
>
> Following the reviewer's suggestion, we have included Noisier2Noise as a baseline in our evaluation (Tables 2 and 3). The results demonstrate the advantages of our approach: on the M4Raw dataset, C2S outperforms Noisier2Noise by approximately 1dB in PSNR across all contrasts (e.g., 32.77dB vs 31.60dB for T1). On the fastMRI dataset, C2S similarly shows better performance.
>
> > W3 (Comparison with Recent Blind-Spot Networks):
>
> We appreciate the reviewer's suggestion to evaluate recent blind-spot methods. Following this recommendation, we have conducted experiments with LG-BPN and PUCA on our datasets. We note that for fair comparison, all baseline methods (N2S, N2V, R2R) in our experiments use U-Net architectures with self-attention mechanisms (DDPM architecture without time conditioning) and carefully tuned hyperparameters (e.g. masking ratios for N2S). Our updated results in Table 2 and 3 show that while LG-BPN and PUCA have demonstrated impressive performance on natural images, their effectiveness on MRI data is more limited. For example on the M4Raw dataset, LG-BPN achieves PSNR/SSIM scores of 31.15/0.890, 30.66/0.868, and 30.82/0.862 for T1, T2, and FLAIR contrasts respectively, while PUCA attains 30.52/0.870, 29.11/0.827, and 29.57/0.807.
> We have updated Tables 2 and 3 to include additional results and provided visual comparisons in Figures 3 and 4 (we observe that the blind-spot architecture design fundamental to both LG-BPN and PUCA often leads to oversmoothing in MRI data potentially due to the limited receptive field).

---

> > ### Author Response · Authors · 2024-11-28
> >
> > > W4 (Visual Comparison with R2R):
> >
> > We appreciate the reviewer's careful observation of the visual results. We acknowledge that in some cases, the primary variant of C2S (with $\sigma_{t_{\text{target}}} = 0$) might lead to oversmoothing. In our updated Figure 4, we present results from both the primary C2S and C2S with detail refinement, which often achieves a better balance between noise reduction and feature preservation.
> >
> > Additionally, we identified that the differences in sharpness between C2S and R2R in the original figures might stem from an implementation detail: while R2R was provided with true noise standard deviations, our C2S results used estimated noise levels. To ensure a fair comparison, we have conducted new experiments where all methods use the same noise information and updated Figure 4 accordingly.
> >
> > > W5 (Fairness of Supervised Method Comparison):
> >
> > We appreciate the reviewer's concern about the comparison with supervised methods. We want to clarify that obtaining completely noise-free "ground truth" images in MRI is fundamentally impossible due to the inherent physics of MR acquisition. Even with multiple averaging, which we use to generate higher-SNR labels for supervised training for the M4Raw dataset, some level of noise inevitably remains. This reflects the real-world challenges in MRI, where acquisition time and cost constraints often limit the achievable SNR.
> >
> > We acknowledge that our statement about "surpassing supervised methods" should be more nuanced. We will revise it to: "This indicates the potential of self-supervised learning to achieve competitive performance with supervised approaches when the latter are trained on practically obtainable higher-SNR labels, particularly in scenarios where perfectly clean ground truth is unavailable, offering a practical and robust solution adaptable to broader clinical settings."
> >
> > > W6 (Fairness of Multi-Contrast Experiments):
> >
> > We thank the reviewer for this important observation. Following the suggestion, we have conducted an extended evaluation where we adapted baseline methods (N2S and R2R) to handle multiple contrasts through input concatenation, similar to our approach. The comprehensive results are now presented in Appendix D (Table 8).
> >
> > The extended experiments show that even with multi-contrast inputs, C2S maintains advantages:
> >
> > 1. For T1 denoising using T1 and T2 contrasts, C2S achieves 33.57dB/0.925 PSNR/SSIM, outperforming the multi-contrast versions of R2R (32.16dB/0.833) and N2S (29.99dB/0.805).
> >
> > 2. Similar improvements are observed across all contrast combinations, with C2S consistently showing 1-3dB PSNR gains over the enhanced baselines.
> >
> > 3. Our analysis reveals that while simple input concatenation allows baselines to access multi-contrast information, they struggle to effectively leverage this complementary information. N2S shows signs of overfitting, while R2R achieves only moderate improvements over its single-contrast version.
> >
> >
> > > W7 (Hallucination):
> >
> > While hallucination is indeed a significant concern in generative diffusion models, we believe that our method is protected against this issue and here we provide two perspectives:
> >
> > 1. Unlike generative diffusion models that learn to synthesize images from pure noise, C2S operates in a discriminative setting where it directly estimates $\mathbb{E}[\mathbf{X}_0 \mid \mathbf{X}_{t_{\text{data}}}]$ (Section 3). This conditional estimation task is inherently constrained by the noisy input, significantly limiting the possibility of hallucination.
> >
> > 2. Our GADSM framework (Theorem 1) ensures that the model learns to estimate the MMSE solution, which by definition minimizes the expected squared error between the prediction and true clean image. This theoretical foundation prevents the model from introducing features not present in the original data. The detail refinement extension (Appendix G) further encourages faithful preservation of anatomical structures.

---

> ### Author Response · Authors · 2024-11-28
>
> > Q1 (Unique Challenges and Importance of MRI Denoising)
>
> MRI denoising presents several distinct challenges that warrant specialized solutions beyond natural image denoising approaches. We have revised our manuscript to better highlight these key differences.
>
> Firstly, MRI data exhibits unique noise characteristics due to the physics of acquisition, with MRI signals following Rician or non-central chi distributions in magnitude images. In medical imaging contexts, visual quality and preservation of diagnostically relevant features are often more critical than standard metrics like PSNR/SSIM. While our method achieves competitive PSNR/SSIM scores, we specifically designed our detail refinement technique to preserve fine features that are crucial for accurate diagnosis but might be sacrificed by methods only focusing on numerical metrics.
>
> A fundamental challenge in MRI is the difficulty in obtaining clean ground truth data due to physical and practical constraints. This limitation motivated our comprehensive comparison between supervised methods (trained on practically obtainable higher-SNR labels) and self-supervised approaches. Our findings in Tables 2 reveal that architectural innovations designed for natural images may not directly translate to improved performance in medical imaging contexts, highlighting the need for specialized approaches. Our approach provides a standalone self-supervised denoising framework that can handle varying noise levels through its score-based formulation.
>
> MRI also offers unique opportunities through multi-contrast acquisition, providing complementary anatomical information not available in natural images. Our framework effectively leverages these structural relationships across different contrasts, as demonstrated by our multi-contrast results in Table 4. The practical impact of these challenges is particularly evident in low-field MRI applications, where noise poses a major barrier to clinical adoption. Our method's robust performance across varying noise conditions and contrasts, combined with its self-supervised nature, makes it valuable for clinical settings where acquiring high-quality training data is challenging.
>
> > Q2 (Multi-contrast Implementation Details):
>
> In our multi-contrast implementation, auxiliary contrasts are used as conditional information. Specifically, given a noisy target contrast image $X_{t_{data}} \in R^d$ and auxiliary contrast images $C \in R^{d \times c}$, where $c$ represents the number of auxiliary contrasts, our model learns to estimate $E[X_0 | X_{t_{data}}, C]$ (Appendix D.2).
>
> The conditioning architecture processes auxiliary contrasts through a CNN encoder to extract features, which are then concatenated with the target contrast features in feature space. These combined features are subsequently processed through a diffusion model backbone with NVC-MSA modules, enabling effective integration of complementary information across contrasts while maintaining the core GADSM framework. The multi-contrast loss function remains consistent with the single-contrast case but incorporates the conditional information:$ L_{MC-C2S}(\theta) = \frac{1}{2} E[w(\tau) \|D_\theta(X_\tau, \tau | C) - X_{t_{data}}\|_2^2]$. Detailed loss functions, implementation specifics, including the complete training algorithm and experimental results demonstrating the effectiveness of this approach, can be found in Appendix D.2. Our experiments show that this conditional approach effectively leverages complementary information across contrasts, leading to marked performance improvements (Table 6).
>
> [1].Nick Moran, Dan Schmidt, Yu Zhong, and Patrick Coady. Noisier2noise: Learning to denoise from unpaired
> noisy data. In Proceedings of the IEEE/CVF Conference on Computer Vision and Pattern Recognition, pp.
> 12064–12072, 2020.
>
> [2].Zichun Wang, Ying Fu, Ji Liu, and Yulun Zhang. Lg-bpn: Local and global blind-patch network for self-
> supervised real-world denoising. In Proceedings of the IEEE/CVF Conference on Computer Vision and
> Pattern Recognition, pp. 18156–18165, 2023.
>
> [3].Hyemi Jang, Junsung Park, Dahuin Jung, Jaihyun Lew, Ho Bae, and Sungroh Yoon. Puca: patch-unshuffle
> and channel attention for enhanced self-supervised image denoising. Advances in Neural Information
> Processing Systems, 36, 2024.

---

> > ### Comment · Reviewer_9qMe · 2024-12-03
> >
> > I appreciate the authors' feedback, explanations, and new experiments. The updated manuscript shows clearer contribution to MRI denoising, with good performance and practicability. But I reserve my point of view that this paper's contribution is limited to applying the ADSM method to MRI denoising with some straightforward adaptations, without innovation in ambient diffusion itself. Therefore, I raised my score to 6.

---

> ### Author Response · Authors · 2024-12-03
>
> We sincerely thank you for the thoughtful feedback and acknowledge your perspective regarding the theoretical contributions. While our work builds upon ADSM, we would like to clarify the key technical contributions of our work. GADSM provides a unified framework that bridges DSM and ADSM by introducing a target noise level parameter. When this target matches the data noise level, it reduces to DSM; when set to zero, it corresponds to ADSM. This generalization enables our detail refinement module to balance noise reduction and preservation of anatomical details crucial for diagnostic purposes. Additionally, to enhance training stability and improve convergence, we introduced a reparameterization strategy that transforms how noise levels are sampled during training. This approach ensures consistent coverage of noise levels during training and makes the process more resilient to noise estimation errors. We appreciate your recognition of our paper's practical contributions and thank you for the increased score.

---

### Official Review · Reviewer_2ysz · 2024-11-04

**Soundness:** 2
**Presentation:** 3
**Contribution:** 2
**Rating:** 5
**Confidence:** 3

**Summary:**

This paper presents Corruption2Self (C2S), a score-based self-supervised denoising framework specifically designed for MRI data. C2S uses a reparameterized noise schedule and applies Generalized Ambient Denoising Score Matching (GADSM) to extend traditional score matching approaches to scenarios where only noisy data are available. Key contributions include the introduction of a reparameterized noise level function to stabilize training, as well as a multi-contrast extension to leverage complementary information across MRI contrasts. Experimental evaluations on M4Raw and fastMRI datasets show competitive results for C2S, often surpassing both classical and self-supervised methods in denoising performance metrics.

**Strengths:**

* Reparameterization of Noise Levels: The proposed reparameterization of noise levels is a noteworthy contribution, offering enhanced training stability and convergence. This allows the model to sample uniformly across the noise range, leading to smoother training curves and better generalization.
* Comparison with Self-Supervised and Supervised Methods: The paper includes extensive quantitative comparisons with self-supervised and supervised denoising models, establishing C2S as a strong self-supervised alternative in terms of PSNR and SSIM.
* Multi-Contrast Extension: Incorporating multi-contrast data is a beneficial approach that leverages complementary MRI contrasts, enhancing structural preservation and improving the quality of denoised images.

**Weaknesses:**

* Limited Novelty Beyond Classical Denoising Diffusion Probabilistic Models (DDPM): The use of a score-based approach is similar to DDPM without substantial differentiation. Although the reparameterization is innovative, the rest of the framework closely resembles classical score-based diffusion models, raising concerns about the originality of the overall approach.
* Noise Level Estimation Error Not Clearly Specified: While Figure 5 attempts to show robustness to noise level estimation error, the specific impacts and handling of these errors in practical settings remain unclear. Further clarity on this aspect would strengthen the model’s practical applicability.
(Fluctuations in Training Stability: Although the reparameterization claims to stabilize training, Figure 2 indicates some fluctuations, suggesting that stabilization might not be consistent across all noise levels. This could impact reproducibility and model robustness, especially under varying noise conditions.
* Parameter Specification in Equations: The notation for key parameters, such as $\lambda_{out}$ ​and $\lambda_{skp}$​ , lacks clear definitions in the methods section, which could hinder understanding and replication of the proposed framework.
* Effects of Corruption Level (T) in Training: The paper does not provide guidance on the relationship between higher corruption levels (T) and the required number of training iterations to achieve optimal performance. This omission could limit the applicability of the method in datasets with different noise characteristics or levels of corruption.
* Counterintuitive Results in Multi-Contrast Experiments: In the multi-contrast experiments, incorporating T1 contrast data seems to worsen the results. This is surprising since T1 typically offers structure-rich information that could enhance denoising. An analysis of this phenomenon would provide deeper insights into the limitations of C2S in multi-contrast applications.

**Questions:**

1. Given that this method is closely aligned with score-based DDPM, what aspects of C2S differentiate it from classical diffusion-based approaches, aside from the reparameterized noise level function?
1. How does the noise level estimation error affect denoising quality in practical scenarios? Is there a threshold or method for mitigating significant deviations in noise estimation?
1. How would the training duration change if a higher maximum corruption level $T$ were used? Does this require additional training epochs to reach convergence, as hinted at in the ablation study?
1. Could the authors clarify why adding T1 contrast data in the multi-contrast experiments led to reduced performance? This is counterintuitive, as T1 contrast generally provides valuable anatomical information.

---

> ### Author Response · Authors · 2024-11-28
>
> Thank you for your valuable and constructive feedback.
> > W1 \& Q1 (Novelty Beyond Classical DDPMs):
>
> While C2S builds upon score-based principles, it introduces several innovations beyond classical DDPMs. First, our Generalized Ambient Denoising Score Matching (GADSM) enables learning directly from noisy data without clean samples -- a key departure from DDPMs which require clean training data. C2S emphasizes a discriminative denoising framework for approximating the MMSE estimator in a self-supervised setting, critical for MRI applications where clean data acquisition is impractical. Second, GADSM lays a foundation for the detail refinement module we proposed, to balance noise reduction and feature preservation, achieving superior diagnostic performance while preserving fine anatomical details crucial for diagnostic purposes. Our proposed score-based self-supervised denoising framework also allows us to take advantage of the recent advances in diffusion model architectures, based on which our model additionally incorporated a Noise Variance Conditioned Multi-Head Self-Attention (NVC-MSA) module, improving performance over classical DDPMs. Lastly, unlike previous self-supervised methods primarily tested on natural images, we provide extensive validation on real clinical MRI data (M4Raw) and simulated datasets (fastMRI), demonstrating robust performance across varying noise conditions and contrasts.
>
> >  W2 \& Q2 (Noise Level Estimation and Training Stability):
>
> We have expanded our analysis of noise level estimation in Appendix H, which now provides comprehensive empirical validation of C2S's robustness to estimation errors. We note that our framework employs scikit-image's noise variance estimation package as a lightweight, practical solution for real-time $\sigma_{t_{\text{data}}}$ estimation. Our experiments demonstrate stable performance with minimal PSNR degradation (maximum variations of only 0.053dB, 0.106dB, and 0.084dB for T1, T2, and FLAIR contrasts respectively) even with ±50\% estimation errors. This robustness enables effective performance when using standard noise estimation tools (e.g., from scikit-image), facilitating practical deployment in clinical settings. Regarding training stability, the observed fluctuations in Figure 2 are a common characteristic of score-based frameworks due to the multi-scale/level noise corruption process[1]. While the multi-scale forward corruption introduces variability through random noise scale sampling, this approach helps enhance denoising performance by enabling better sampling in low-density regions, leading to more accurate score estimations. To further improve stability, we incorporated Exponential Moving Average (EMA) of model parameters on top of our reparameterization technique. As shown in the updated Figure 2, this combination provides notably smoother convergence compared to the baseline. Despite the inherent fluctuations in multi-scale score-based training, our method demonstrates consistent and reproducible performance across multiple training runs and datasets.
>
> > W3 (Parameter Specification):
>
> We have enhanced our manuscript by revising Section 3.2 to provide explicit definitions and interpretations for all key parameters, including ($\lambda_{out}$) and ($\lambda_{skip}$), upon their first introduction.
>
> [1].Yang Song and Stefano Ermon. Improved techniques for training score-based generative models. Advances in neural information processing systems, 33:12438–12448, 2020.

---

> > ### Author Response · Authors · 2024-11-28
> >
> > > W4 \& Q3 (Effects of Corruption Level T):
> >
> > We have analyzed the relationship between the maximum corruption level T and training dynamics, with detailed results now presented in Appendix I. As demonstrated in Table 12, increasing T leads to approximately linear growth in required training epochs, ranging from 42 epochs at T=3 to 204 epochs at T=20. Increasing T beyond optimal values provides diminishing returns while unproportionally increasing computational costs due to the exposure of the model to a broader range of noise levels. Based on these findings, we have added practical guidelines in Appendix I to help practitioners select appropriate T values and training durations.
> >
> > > W5 \& Q4 (Multi-Contrast Performance):
> >
> > We thank the reviewer for this insightful observation. Upon investigation, we found that our previous results were affected by incomplete model convergence. After further fine-tuning, our updated results demonstrate that incorporating T1 contrast consistently improves denoising performance across all contrasts. For T2 denoising, T1+T2 (33.11dB/0.901) demonstrate substantial gains over the baseline. Similarly, for FLAIR denoising, incorporating T1 yields 32.98dB/0.901. These results confirm that T1's structure-rich information indeed enhances denoising performance when properly leveraged. We have updated the manuscript with quantitative results (Table 6) and visual comparisons (Figure 6 in Appendix D), demonstrating the improved structural detail preservation achieved by multi-contrast C2S.

---

> ### Author Response · Authors · 2024-12-04
>
> We are thankful for your detailed review and valuable feedback. Since the discussion period ends soon, we would like to kindly check whether we have addressed all of your concerns and we are happy to address any remaining questions. Thank you again and we look forward to your reply.

---

### Author Response · Authors · 2024-12-02
**General response**

We sincerely thank all reviewers for their dedicated time and thorough evaluation of our work. We are encouraged that reviewers recognize the following merits of this paper:

- The mathematical foundation and methodology are sound (Reviewer `9qMe`), with comprehensive analysis and great detail in the workflow (Reviewer `jWG5`)

- The proposed reparameterization of noise levels is a noteworthy contribution, enhancing training stability and convergence (Reviewer `2ysz`, `jWG5`)

- The experimental validation is extensive and robust, including:
  - Comprehensive comparisons with both self-supervised and supervised methods (Reviewer `2ysz`)
  - Thorough experiments demonstrating superior performance against various baselines (Reviewer `9qMe`)
  - Validation of robustness across varied noise level estimations (Reviewer `jWG5`)

- The extension to multi-contrast MRI is valuable and novel:
  - Effectively leverages complementary MRI contrasts for enhanced structural preservation (Reviewer `2ysz`)
  - Addresses an overlooked field in MRI denoising (Reviewer `9qMe`)


We have carefully addressed all the raised concerns and questions in our detailed responses below. The paper has been revised with major modifications highlighted in blue color, and we have added new analysis and content encouraged by the reviewers' insightful comments.

We deeply appreciate the time and effort all reviewers have dedicated to providing such constructive feedback, which has helped us significantly improve our manuscript. With the discussion phase ending on December 2nd, we warmly welcome any further comments or suggestions.

---

### Meta-Review · Area_Chair_1CuE · 2024-12-23

**Metareview:**

This paper introduces Corruption2Self (C2S), a self-supervised denoising framework for MRI data that leverages a reparameterized noise schedule and Generalized Ambient Denoising Score Matching (GADSM) to handle noisy data without reliance on clean labels. The framework includes a reparameterized noise level function for stable training, a multi-contrast extension to integrate complementary MRI contrasts, and detail refinement for preserving finer spatial features. Experimental results on multiple datasets show that C2S outperforms previous self-supervised methods and achieves competitive results compared to supervised approaches.

Strength: The reparameterization of noise levels enhances training stability and convergence, leading to better generalization. Incorporating multi-contrast MRI data improves structural preservation and denoising quality. Extensive comparisons with supervised and self-supervised methods support method effectiveness.

Weakness: The reviewers raise the concerns about the limited contribution of the method, which is a straightforward extension of existing techniques ADSM for MRI denoising, limited baseline methods comparison, and parameter clarification. The author has provided a detailed response including further explanations and new experiments to address these concerns.

**Additional Comments On Reviewer Discussion:**

During the rebuttal process, the authors provided detailed responses to the reviewers' comments, effectively addressing most of the concerns. Two out of three reviewers engaged with the rebuttal, with one increasing their score to reflect the improvements made in both the manuscript and the authors' responses. Following the rebuttal, the overall scores converged to above borderline, with the only negative score coming from a reviewer who did not respond further. After reviewing the paper, the reviewers' feedback, and the rebuttals, I believe the authors have adequately addressed the raised concerns.

---

### Decision · Program_Chairs · 2025-01-22

Accept (Poster)